# Appropriate relaxation of non-pharmaceutical interventions minimizes the risk of a resurgence in SARS-CoV-2 infections in spite of the Delta variant

**Wadim Koslow**[1], **Martin J. Kühn**[1,2], **Sebastian Binder**[2], **Margrit Klitz**[1], **Daniel Abele**[1], **Achim Basermann**[1]*, **Michael Meyer-Hermann**[2]*

**1** Institute for Software Technology, Department of High-Performance Computing, German Aerospace Center, Cologne, Germany, **2** Department of Systems Immunology and Braunschweig Integrated Centre of Systems Biology (BRICS), Helmholtz Centre for Infection Research, Braunschweig, Germany

* achim.basermann@dlr.de (AB); mmh@theoretical-biology.de (MM-H)

**Data Availability Statement:** All parameters to setup the simulations are provided in the current article. The whole simulation code and the tools to

## Abstract

We analyze the relaxation of non-pharmaceutical interventions (NPIs) under an increasing number of vaccinations in Germany. For the spread of SARS-CoV-2 we employ a SIR-type model that accounts for age-dependence and includes realistic contact patterns between age groups. The implementation of NPIs occurs on changed contact patterns, improved isolation, or reduced infectiousness when, e.g., wearing masks. We account for spatial heterogeneity and commuting activities in between regions in Germany, and the testing of commuters is considered as a further NPI. We include the ongoing vaccination process and analyze the effect of the B.1.617.2 (Delta) variant, which is considered to be 40%–60% more infectious then the currently dominant B.1.1.7 (Alpha) variant. We explore different opening scenarios under the ongoing vaccination process by assuming that local restrictions are either lifted in early July or August with or without continued wearing of masks and testing. Our results indicate that we can counteract the resurgence of SARS-CoV-2 despite the Delta variant with appropriate timing for the relaxation of NPIs. In all cases, however, school children are hit the hardest.

## Author summary

One of the greatest challenges within the Covid-19 pandemic is to identify the timing and amount of non-pharmaceutical interventions (face masks, travel bans, school closures, etc). In the year 2021 more and more people are getting vaccinated. When can we finally lift all restrictions and stop wearing masks? In order to provide more insights to this question, we use a mathematical model which is capable of simulating the effects of non-pharmaceutical interventions in Germany while accounting for age-dependent factors as well as commuting activities between regions. We include the vaccination process and analyze the much more infectious Delta coronavirus variant. We simulate scenarios that consider the timing of the return to pre-pandemic contacts as well as when to suspend wearing

gather the input data are available under an open source software at: https://github.com/DLR-SC/memilio.

**Funding:** S.B. has received funding from the European Union's Horizon 2020 research and innovation programme under grant agreement No 101003480. S.B. has received funding from the Initiative and Networking Fund of the Helmholtz Association. S.B. was supported by German Federal Ministry of Education and Research for the project CoViDec (FKZ: 01KI20102). M.J.K. acknowledges the Helmholtz Information & Data Science Academy (HIDA) for providing financial support within the HIDA Trainee Network program enabling a short-term research stay at Department of Systems Immunology and Braunschweig Integrated Centre of Systems Biology (BRICS), Helmholtz Centre for Infection Research, Braunschweig to analyze data sets that were used as input data for our models. The funders had no role in study design, data collection and analysis, decision to publish, or preparation of the manuscript.

**Competing interests:** The authors have declared that no competing interests exist.

masks and testing. Our results show that a later opening by 1 August in combination with masks and testing reduces the chance of a further infection wave considerably. From the retrospective view of the revision, we see that the rise in infections at the end of summer could have been well predicted by our scenarios that considered lifting of NPIs in July as it happened in many places. In all of our scenarios, the infection manifests in the younger age groups.

## Introduction

After almost two years, the coronavirus disease 2019 (Covid-19) continues to have a tremendous impact on daily life in many countries. Even though the vaccination process had been rapidly progressing in Germany over the summer, herd immunity is still far from being achieved [1]. It still remains questionable whether the vaccination readiness of the population is sufficiently high to reach the herd immunity threshold [2] in the near future. However, maintaining compliance with restrictions within the population becomes more challenging as the duration of the pandemic increases, e.g. the risk perception did not vary systematically with case numbers in April 2021 in Germany [3] suggesting strong habituation effects. With a decreasing incidence in summer, relaxation of measures were inevitable and desirable to minimize economic and social costs. Hence, a cautious relaxation of measures in lockstep with increasing vaccination success is generally considered advisable [2, 4, 5].

The aim of this paper is to simulate different NPI relaxation strategies over the summer and to consider different restrictions with the upcoming winter to analyze their consequences while the number of vaccinations continues to rise. We specifically investigate the effect on the younger age groups. To this end, we employ our previously developed SIR-type model [6]. In this model, we account for the age-dependence of the severe acute respiratory syndrome Coronavirus 2 (SARS-CoV-2) and include realistic contact patterns between age groups. The implementation of NPIs occurs on changed contact patterns, improved isolation, or reduced infectiousness when, e.g., wearing masks. In order to account for spatial heterogeneity, we use a graph approach and we include high-quality information on commuting activities combined with traveling information from social networks. We expand our model by new compartments to represent the dynamics of the ongoing vaccination process and even allow for reinfections or infections after full vaccination. Additionally, we implement the effect of the B.1.617.2 (Delta) Coronavirus variant, which is considered to be 40%–60% more infectious then the previously dominant B.1.1.7 (Alpha) variant; cf. [7–9]. In this paper, we focus on the comparison of different strategies based on our mathematical model that was already validated in [6]. From the retrospective view of the revision, we see that the rise in infections could have been well predicted.

Only few studies analyze the effect of NPIs during the ongoing vaccination process while also considering age-stratification as well as spatial heterogeneity. Patel et al. [10] introduce an agent based SEIR-type model for North Carolina which simulates different vaccine coverage and efficacy scenarios with NPIs without spatial heterogeneity. Moore et al. use an approach similar to ours for the UK [2, 11]. The focus of these studies is on the vaccine efficacy and coverage without distinguishing between different types of NPIs. Bauer et al. [4] and Viana et al. [12] examine NPI relaxation strategies without spatial heterogeneity in the EU and Portugal, respectively. All of these studies are tailored to their specific region, and to our knowledge no such study exists for Germany. Here, Maier et al. [13] discuss the benefits of delaying the second dose of the vaccine without specific focus on NPIs, and the comment [5] generally stresses

the lift of restrictions in pace with vaccination. The authors of [14] considered different vaccination ratios to avoid further rising of infections in autumn and winter. In this paper we specifically want to look at the effect of the timing of NPI release, as well as the subsequent effect of masks while accounting for the impact of school holidays and more contagious variants like the Delta variant in the different age groups. In the revision, we also added different scenarios for the upcoming winter period.

The paper is structured as follows: First, we introduce the mathematical model. Then, we define scenarios with different relaxation strategies and NPIs. Finally, we present the simulation results and discuss their implications for decisions on NPI relaxations or restrictions.

## Materials and methods

The methods used in this paper are based on mathematical models. The model is based on our previously developed hybrid graph-SIR-type or metapopulation model in [6, 15], extended by partial and full vaccination as well as infection after immunization following infection or vaccination. In order to sufficiently represent reality with our compartmental model, we use two key ingredients: Age stratification and spatial resolution. Both will be explained in the following paragraphs.

Our original model consists of the following compartments: *Susceptible* (S), healthy individuals without immune memory of SARS-CoV-2; *Exposed* (E), who carry the virus but are not yet infectious to others; *Carrier* (C), who carry the virus and are infectious to others but do not yet show symptoms (they may be pre- or asymptomatic); *Infected* (I), who carry the virus, are infectious and show symptoms; *Hospitalized* (H), who experience a severe development of the disease; *In Intensive Care Unit* (U); *Dead* (D); and *Recovered* (R), who could not be infected again in our original model.

In order to model commuter testing, we use the compartments $C^+$ and $I^+$ from [15] for carriers or infected individuals as well as their (partially) vaccinated counterparts $C_{PV}^+$, $I_{PV}^+$, $C_V^+$ and $I_V^+$ In what follows, we will provide details on both the vaccination model as well as on the details of commuter testing.

### SIR-type model with vaccination

We expand the model by the compartments of *Partially Vaccinated Susceptible* ($S_{PV}$), individuals that have received the first dose of the vaccine; *Partially Vaccinated Exposed* ($E_{PV}$), who carry the virus despite being partially vaccinated but are not yet infectious to others as well as *Partially Vaccinated Carrier* ($C_{PV}$); *Partially Vaccinated Infected* ($I_{PV}$); *Partially Vaccinated Hospitalized* ($H_{PV}$); *Partially Vaccinated In Intensive Care Unit* ($U_{PV}$). Furthermore, we introduce these compartments for fully vaccinated individuals as $S_V$, $E_V$, $C_V$, $I_V$, and $H_V$. Additionally, the compartment that we previously defined as "Recovered" will now be denoted as "Immune" to also represent vaccinated individuals.

To account for the vaccination, we make a number of simplifying modeling assumptions. We equate fully vaccinated individuals one week after the second dose to those who gained immunity by recovering from Covid-19. In the literature different vaccines show different effectiveness after about one week of administering the second dose [16, 17], and we also refer to [18] showing an antibody peak about 6–8 days after the second dose.

We here provide a model where partially vaccinated as well as recovered or fully vaccinated people can get infected, be infectious to others or experience a severe course of infection [16, 17]. We assume the partial vaccination to take effect after two weeks after the first dose.

We will explain the modeling of vaccine efficacy and provide references for the chosen parameter ranges in Eq (30) and thereafter.

**Age-stratification and full local model.** The model is visualized without age groups in Fig 1. To resolve age-specific disease parameters, we divide the totality of people $N$ into $i = 1,$ $\ldots, n = 6$ different age groups as defined in Table 1. Thus, all of our compartments have an age-dependence that we indicate by the subscript $i$. We define $\mathcal{Z}_i := \{S_i, E_i, C_i, \ldots\}$ as the set of all compartments of age group $i$ and $N_i^{D^\perp} := \sum_{z \in \mathcal{Z}_i \setminus D_i} z$ as the sum of all living individuals of age group i.

We use the variables $T_{*^1}^{*^2}$ for the time spent in state $*^1 \in \mathcal{Z}_i$ before transition to state $*^2 \in \mathcal{Z}_i$ and $\mu_{*^1}^{*^2}$ for the probability of a patient to go to state $*^2$ from state $*^1$.

We write the whole systems of equations as

$$\frac{dS_i}{dt} = -S_i \rho_i \sum_{j=1}^{n} \phi_{i,j} \frac{\xi_{C,j}(C_j + C_{PV,j} + C_{V,j}) + \xi_{I,j}(I_j + I_{PV,j} + I_{V,j})}{N_j^{D^\perp}}, \tag{1}$$

$$\frac{dE_i}{dt} = S_i \rho_i \sum_{j=1}^{n} \phi_{i,j} \frac{\xi_{C,j}(C_j + C_{PV,j} + C_{V,j}) + \xi_{I,j}(I_j + I_{PV,j} + I_{V,j})}{N_j^{D^\perp}} - \frac{1}{T_{E_i}^{C_i}} E_i, \tag{2}$$

$$\frac{dC_i}{dt} = \frac{1}{T_{E_i}^{C_i}} E_i - \left( \frac{1 - \mu_{C_i}^{R_i}}{T_{C_i}^{I_i}} + \frac{\mu_{C_i}^{R_i}}{T_{C_i}^{R_i}} \right) C_i, \tag{3}$$

$$\frac{dC_i^+}{dt} = - \left( \frac{1 - \mu_{C_i}^{R_i}}{T_{C_i}^{I_i}} + \frac{\mu_{C_i}^{R_i}}{T_{C_i}^{R_i}} \right) C_i^+, \tag{4}$$

$$\frac{dI_i}{dt} = \frac{1 - \mu_{C_i}^{R_i}}{T_{C_i}^{I_i}} C_i - \left( \frac{1 - \mu_{I_i}^{H_i}}{T_{I_i}^{R_i}} + \frac{\mu_{I_i}^{H_i}}{T_{I_i}^{H_i}} \right) I_i, \tag{5}$$

$$\frac{dI_i^+}{dt} = \frac{1 - \mu_{C_i}^{R_i}}{T_{C_i}^{I_i}} C_i^+ - \left( \frac{1 - \mu_{I_i}^{H_i}}{T_{I_i}^{R_i}} + \frac{\mu_{I_i}^{H_i}}{T_{I_i}^{H_i}} \right) I_i^+, \tag{6}$$

$$\frac{dH_i}{dt} = \frac{\mu_{I_i}^{H_i}}{T_{I_i}^{H_i}} I_i + \frac{\mu_{I_i}^{H_i}}{T_{I_i}^{H_i}} I_i^+ - \left( \frac{1 - \mu_{H_i}^{U_i}}{T_{H_i}^{R_i}} + \frac{\mu_{H_i}^{U_i}}{T_{H_i}^{U_i}} \right) H_i, \tag{7}$$

$$\frac{dU_i}{dt} = \frac{\mu_{H_i}^{U_i}}{T_{H_i}^{U_i}} H_i - \left( \frac{1 - \mu_{U_i}^{D_i}}{T_{U_i}^{R_i}} + \frac{\mu_{U_i}^{D_i}}{T_{U_i}^{D_i}} \right) U_i, \tag{8}$$

$$\frac{dS_{PV,i}}{dt} = -S_{PV,i} \rho_{PV,i} \sum_{j=1}^{n} \phi_{i,j} \frac{\xi_{C,j}(C_j + C_{PV,j} + C_{V,j}) + \xi_{I,j}(I_j + I_{PV,j} + I_{V,j})}{N_j^{D^\perp}}, \tag{9}$$

$$\frac{dE_{PV,i}}{dt} = S_{PV,i} \rho_{PV,i} \sum_{j=1}^{n} \phi_{i,j} \frac{\xi_{C,j}(C_j + C_{PV,j} + C_{V,j}) + \xi_{I,j}(I_j + I_{PV,j} + I_{V,j})}{N_j^{D^\perp}} - \frac{1}{T_{E_i}^{C_i}} E_{PV,i}, \tag{10}$$

$$\frac{dC_{PV,i}}{dt} = \frac{1}{T_{E_i}^{C_i}} E_{PV,i} - \left( \frac{1 - \mu_{C_{PV,i}}^{R_i}}{T_{C_i}^{I_i}} + \frac{\mu_{C_{PV,i}}^{R_i}}{\kappa T_{C_i}^{R_i}} \right) C_{PV,i}, \tag{11}$$

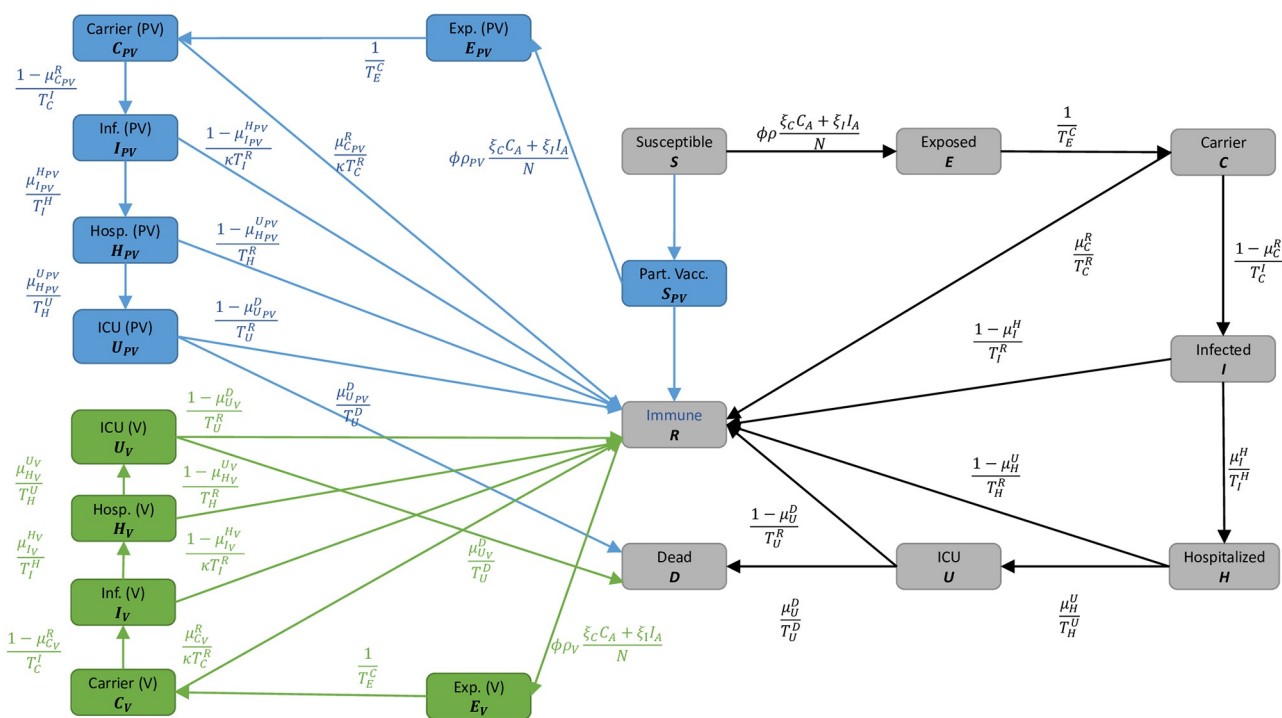

**Fig 1. Local SIR-type model with vaccinations.** We omit the age-dependence index $i$ as well as the compartments $C^+$ and $I^+$ which do not have inflow from other compartments. The blue and green boxes represent compartments that have been newly added for the vaccination model. Here, $C_A = C + C_{PV} + C_V$ and $I_A = I + I_{PV} + I_V$.

$$\frac{dC_{PV,i}^+}{dt} = -\left(\frac{1 - \mu_{C_{PV,i}}^{R_i}}{T_{C_i}^{I_i}} + \frac{\mu_{C_{PV,i}}^{R_i}}{\kappa T_{C_i}^{R_i}}\right)C_{PV,i}^+, \tag{12}$$

$$\frac{dI_{PV,i}}{dt} = \frac{1 - \mu_{C_{PV,i}}^{R_i}}{T_{C_i}^{I_i}}C_{PV,i} - \left(\frac{1 - \mu_{I_{PV,i}}^{H_{PV,i}}}{\kappa T_{I_i}^{R_i}} + \frac{\mu_{I_{PV,i}}^{H_{PV,i}}}{T_{I_i}^{H_i}}\right)I_{PV,i}, \tag{13}$$

$$\frac{dI_{PV,i}^+}{dt} = \frac{1 - \mu_{C_{PV,i}}^{R_i}}{T_{C_i}^{I_i}}C_{PV,i}^+ - \left(\frac{1 - \mu_{I_{PV,i}}^{H_{PV,i}}}{\kappa T_{I_i}^{R_i}} + \frac{\mu_{I_{PV,i}}^{H_{PV,i}}}{T_{I_i}^{H_i}}\right)I_{PV,i}^+, \tag{14}$$

$$\frac{dH_{PV,i}}{dt} = \frac{\mu_{I_{PV,i}}^{H_{PV,i}}}{T_{I_i}^{H_i}}I_{PV,i} + \frac{\mu_{I_{PV,i}}^{H_{PV,i}}}{T_{I_i}^{H_i}}I_{PV,i}^+ - \left(\frac{1 - \mu_{H_{PV,i}}^{U_{PV,i}}}{T_{H_i}^{R_i}} + \frac{\mu_{H_{PV,i}}^{U_{PV,i}}}{T_{H_i}^{U_i}}\right)H_{PV,i}, \tag{15}$$

$$\frac{dU_{PV,i}}{dt} = \frac{\mu_{H_{PV,i}}^{U_{PV,i}}}{T_{H_i}^{U_i}}H_{PV,i} - \left(\frac{1 - \mu_{U_{PV}i}^{D_i}}{T_{U_i}^{R_i}} + \frac{\mu_{U_{PV,i}}^{D_i}}{T_{U_i}^{D_i}}\right)U_{PV,i}, \tag{16}$$

$$\frac{dS_{V,i}}{dt} = -S_{V,i}\rho_{V,i}\sum_{j=1}^{n}\phi_{i,j}\frac{\xi_{C,j}(C_j + C_{PV,j} + C_{V,j}) + \xi_{I,j}(I_j + I_{PV,j} + I_{V,j})}{N_j^{D\perp}}, \tag{17}$$

**Table 1. Parameter list for the local SIR-type model.** Parameter list for the local SIR-type model as presented in Fig 1. For details on parameter estimations of the base model, see [6].

| param. | range in age group | | | | | |
|---|---|---|---|---|---|---|
| | **0–4** | **5–14** | **15–34** | **35–59** | **60–79** | **80+** |
| $\rho_i^{(0)}$ | [0.028, 0.056] | [0.070, 0.098] | | | [0.11, 0.14] | [0.14, 0.21] |
| $k$ | [0.1, 0.3] | | | | | |
| $\sigma_\delta$ | see (35) | | | | | |
| $\nu_\delta$ | 1.4 or 1.6 | | | | | |
| $\xi_{C,i}$ | sigmoidal curve from 0.5 to 1 on incidence 10 to 20 | | | | | |
| $\xi_{I,i}$ | sigmoidal curve from [0.0, 0.2] to [0.4, 0.5] on incidence 10 to 150 | | | | | |
| $T_E^C$ | [2.67, 4.00] | | | | | |
| $\mu_C^R$ | [0.20,0.30] | [0.15,0.25] | | | | |
| $T_C^R$ | $T_C^I + 0.5 T_I^R$ | | | | | |
| $T_C^I$ | sampled with $T_E^C : T_E^C + T_C^I = 5.2$ | | | | | |
| $\mu_I^H$ | [0.006,0.009] | [0.015,0.023] | [0.049,0.074] | | [0.15,0.18] | [0.20,0.25] |
| $T_I^H$ | [9.0,12.0] | | | | [5.0,7.0] | |
| $T_I^R$ | [5.6,8.4] | | | | | |
| $\mu_H^U$ | [0.05,0.10] | | [0.10,0.20] | | [0.25,0.35] | [0.35,0.45] |
| $T_H^U$ | [3.0,7.0] | | | | | |
| $T_H^R$ | [4.0,6.0] | [5.0,7.0] | [7.0,9.0] | | [9.0,11.0] | [13.0,17.0] |
| $\mu_U^D$ | [0.00,0.10] | [0.10,0.18] | | | [0.3,0.5] | [0.5,0.7] |
| $T_U^R$ | [5.0,9.0] | | [14.0,21.0] | | | [10.0,15.0] |
| $T_U^D$ | [4.0,8.0] | | [15.0,18.0] | | | [10.0,12.0] |
| $\kappa$ | 0.5 or 1 | | | | | |
| $p_{E_{PV}}$ | [0.15, 0.25] | | | | | |
| $p_{I_{PV}}$ | [0.3, 0.4] | | | | | |
| $p_{H_{PV}}$ | [0.85, 0.95] | | | | | |
| $p_{U_{PV}}$ | equal to $p_{H_{PV}}$ | | | | | |
| $p_{D_{PV}}$ | equal to $p_{H_{PV}}$ | | | | | |
| $p_{E_V}$ | [0.619, 0.719] | | | | | |
| $p_{I_V}$ | [0.707, 0.807] | | | | | |
| $p_{H_V}$ | [0.859, 0.959] | | | | | |
| $p_{U_V}$ | equal to $p_{H_V}$ | | | | | |
| $p_{D_V}$ | equal to $p_{H_V}$ | | | | | |
| $T_{PV}^V$ | 49 | | | | | |

$$\frac{dE_{V,i}}{dt} = S_{V,i}\rho_{V,i} \sum_{j=1}^{n} \phi_{i,j} \frac{\xi_{C,j}(C_j + C_{PV,j} + C_{V,j}) + \xi_{I,j}(I_j + I_{PV,j} + I_{V,j})}{N_j^{D^\perp}} - \frac{1}{T_{E_i}^{C_i}} E_{V,i}, \qquad (18)$$

$$\frac{dC_{V,i}}{dt} = \frac{1}{T_{E_i}^{C_i}} E_{V,i} \left( \frac{1 - \mu_{C_V,i}^{R_i}}{T_{C_i}^{I_i}} + \frac{\mu_{C_V,i}^{R_i}}{\kappa T_{C_i}^{R_i}} \right) C_{V,i}, \qquad (19)$$

$$\frac{dC_{V,i}^+}{dt} = - \left( \frac{1 - \mu_{C_V,i}^{R_i}}{T_{C_i}^{I_i}} + \frac{\mu_{C_V,i}^{R_i}}{\kappa T_{C_i}^{R_i}} \right) C_{V,i}^+, \qquad (20)$$

$$\frac{dI_{V,i}}{dt} = \frac{1 - \mu_{C_V,i}^{R_i}}{T_{C_i}^{I_i}} C_{V,i} - \left( \frac{1 - \mu_{I_V,i}^{H_V,i}}{\kappa T_{I_i}^{R_i}} + \frac{\mu_{I_V,i}^{H_V,i}}{T_{I_i}^{H_i}} \right) I_{V,i}, \tag{21}$$

$$\frac{dI_{V,i}^+}{dt} = \frac{1 - \mu_{C_V,i}^{R_i}}{T_{C_i}^{I_i}} C_{V,i}^+ - \left( \frac{1 - \mu_{I_V,i}^{H_V,i}}{\kappa T_{I_i}^{R_i}} + \frac{\mu_{I_V,i}^{H_V,i}}{T_{I_i}^{H_i}} \right) I_{V,i}^+, \tag{22}$$

$$\frac{dH_{V,i}}{dt} = \frac{\mu_{I_V,i}^{H_V,i}}{T_{I_i}^{H_i}} I_{V,i} + \frac{\mu_{I_V,i}^{H_V,i}}{T_{I_i}^{H_i}} I_{V,i}^+ - \left( \frac{1 - \mu_{H_V,i}^{U_V,i}}{T_{H_i}^{R_i}} + \frac{\mu_{H_V,i}^{U_V,i}}{T_{H_i}^{U_i}} \right) H_{V,i}, \tag{23}$$

$$\frac{dU_{V,i}}{dt} = \frac{\mu_{H_V,i}^{U_V,i}}{T_{H_i}^{U_i}} H_{V,i} - \left( \frac{1 - \mu_{U_V,i}^{D_i}}{T_{U_i}^{R_i}} + \frac{\mu_{U_V,i}^{D_i}}{T_{U_i}^{D_i}} \right) U_{V,i}, \tag{24}$$

$$\begin{aligned} \frac{dR_i}{dt} \quad &= \frac{\mu_{C_i}^{R_i}}{T_{C_i}^{R_i}}(C_i + C_i^+) + \frac{1 - \mu_{I_i}^{H_i}}{T_{I_i}^{R_i}}(I_i + I_i^+) + \frac{1 - \mu_{H_i}^{U_i}}{T_{H_i}^{R_i}} H_i + \frac{1 - \mu_{U_i}^{D_i}}{T_{U_i}^{R_i}} U_i, \\ &+ \frac{\mu_{C_{PV},i}^{R_i}}{\kappa T_{C_i}^{R_i}}(C_{PV,i} + C_{PV,i}^+) + \frac{1 - \mu_{I_{PV},i}^{H_{PV},i}}{\kappa T_{I_i}^{R_i}}(I_{PV,i} + I_{PV,i}^+) \\ &+ \frac{1 - \mu_{H_{PV},i}^{U_{PV},i}}{T_{H_i}^{R_i}} H_{PV,i} + \frac{1 - \mu_{U_{PV},i}^{D_i}}{T_{U_i}^{R_i}} U_{PV,i} \\ &+ \frac{\mu_{C_V,i}^{R_i}}{\kappa T_{C_i}^{R_i}}(C_{V,i} + C_{V,i}^+) + \frac{1 - \mu_{I_V,i}^{H_V,i}}{\kappa T_{I_i}^{R_i}}(I_{V,i} + I_{V,i}^+) \\ &+ \frac{1 - \mu_{H_V,i}^{U_V,i}}{T_{H_i}^{R_i}} H_{V,i} + \frac{1 - \mu_{U_V,i}^{D_i}}{T_{U_i}^{R_i}} U_{V,i}, \end{aligned} \tag{25}$$

$$\frac{dD_i}{dt} \quad = \frac{\mu_{U_i}^{D_i}}{T_{U_i}^{D_i}} U_i + \frac{\mu_{U_{PV},i}^{D_i}}{T_{U_i}^{D_i}} U_{PV,i} + \frac{\mu_{U_V,i}^{D_i}}{T_{U_i}^{D_i}} U_{V,i}. \tag{26}$$

Our parameter estimates are essentially based on the parameter ranges and age groups as gathered and described elaborately in [6], Table 1 & 2]. Here, the transmission risk $\rho_i$ has changed due to the Alpha and Delta variant. To make this paper self-contained, we provide all parameter values and ranges in Table 1. For a description of the parameters, see Table 2.

The transmission risk $\rho_i = \rho_i(t)$ depends on the base transmission risk $\rho_i^{(0)}$ given in Table 1 and the seasonality $s_k(t)$ in Eq (28), and we define

$$\rho_i(t) = s_k(t)\rho_i^{(0)}, \quad i \in \{1, \ldots, 6\}. \tag{27}$$

In [6], we had to assume a larger transmission risk for age group 80+ than initially assumed to model difficult transmission dynamics in nursing homes. Given vaccination progress, we could now reassume the initially assumed transmission risk, corrected for Alpha as $\rho_6^{(0)} \in [0.14, 0.21]$. Note that we include the share of the Delta variant later on by increasing the transmission risk over time in Eq (36).

**Table 2. New parameters for the local SIR-type model.** Description of new or modified (in comparison to [6]) parameters.

| parameter | description |
|---|---|
| $\rho_i^{(0)}$ | base transmission risk of the Alpha variant; already used in [15]; adaptation for the Delta variant happens in (36) |
| $\sigma_\delta$ | share of the Delta variant from June 06, 2021, on; cf. (35) |
| $\nu_\delta$ | relative transmission risk of Delta compared to Alpha variant |
| $\xi_{C,i}$ | nondetection and nonisolation of carrier; already used in [15] |
| $\xi_{I,i}$ | nondetection and nonisolation of infected; already used in [15] |
| $\kappa$ | reduction factor for time spans of mild infections of vaccinated individuals |
| $p_{E_{PV}}$ | effectiveness of partial vaccination against asymptomatic infection |
| $p_{I_{PV}}$ | effectiveness of partial vaccination against symptomatic infection |
| $p_{H_{PV}}$ | effectiveness of partial vaccination against hospitalization |
| $p_{U_{PV}}$ | effectiveness of partial vaccination against ICU treatment |
| $p_{D_{PV}}$ | effectiveness of partial vaccination against death |
| $p_{E_V}$ | effectiveness of full vaccination against asymptomatic infection |
| $p_{I_V}$ | effectiveness of full vaccination against symptomatic infection |
| $p_{H_V}$ | effectiveness of full vaccination against hospitalization |
| $p_{U_V}$ | effectiveness of full vaccination against ICU treatment |
| $p_{D_V}$ | effectiveness of full vaccination against death |
| $T_{PV}^V$ | averaged time between first and second vaccination dose |

We keep the seasonality factor as established in [6], namely

$$s_k(t) := 1 + k \sin\left( \pi \left( \frac{t}{182.5} + \frac{1}{2} \right) \right), \tag{28}$$

which adjusts the base transmission risk in Eq (27), where $t$ is the day of the year and $k \in [0.1, 0.3]$. The chosen parameter $k$ will yield scenarios with modest seasonal influences, i.e. a relative reduction of 18.2%–46.2% in transmissibility between winter an summer. With seasonality, we account for increased outdoor contacts during the summer, as opposed to more indoor contacts during the winter and other epidemiological factors regarding seasonality. A recent study [19] estimates a slightly higher effect with a CI of 25%–53%. However, note that per-country comparisons of transmissibility between seasons as seen by statistical models in [19] may disregard behavioral and other changes over the course of the pandemic. Another recent study [4] for Germany uses similar values as we do here.

**Age-stratified contact patterns and NPI implementation.** For the sake of completeness, we briefly rephrase how age-resolved and time-dependent contact patterns are obtained. A baseline contact pattern for the prepandemic phase is obtained from [20, 21]. As in [15], we set the minimum contact pattern introduced in [6] to zero. For details, see [6, Sec. 2.2] and [15]. The contact pattern between age group $i$ and $j$ is then denoted by $\phi_{i,j}$ and appears in (1), (2), (9), (10), (17), and (18). The contact frequency matrix $\Phi = (\phi_{i,j})_{i,j=1,\ldots,n}$ represents all (mean) daily contacts of a person of age groups $i$ with people from age groups $j$. These contact patterns are time-dependent and change according to the interventions (NPIs) in place. Let us denote the baseline number of contacts in the four locations of contact, *home*, *school*, *work*, and *other*, by $\phi_{B,*,i,j}$, $* \in \{H, S, W, O\}$. The resulting number of contacts according to the NPIs then reads

$$\phi_{i,j} = \sum_{* \in \{H,S,W,O\}} \phi_{B,*,i,j} \prod_{l=1}^{2} (1 - r_{*,i,j}^{(l)}). \tag{29}$$

Here, $r_*^{(l)} \in [0, 1]$ is the reduction factor in effective contacts as induced from NPIs. The super-index $l$ is the intervention level. With $l = 1$ we describe interventions that yield direct contact reduction such as gathering bans. With $l = 2$ we include protective effects from, e.g., face masks and distancing. In our simulations, wearing masks equates to an averaged 25–35% reduction in contacts in the categories school, work and other. There is a number of studies indicating that the reduction in infection spread by mask wearing is likely in this range. First, a randomized control trial in Denmark [22] comes to the conclusion that wearing masks reduces the infection risk for the wearer by 18%. By design, this study only measured protection for the wearer and the protective effect for transmission to others can be assumed to be at least as high. Comparative studies for different regions of Germany observed reduced infections between 15% and 75% over a period of 20 days after mandatory introduction of masks [23]. A recent systematic review [24] reports a pooled reduction in Covid-19 incidence by 53% for face masks, with substantial heterogeneity across the studies, however. Another recent review [25] reports values from 15–40% in the discussed studies.

Besides static interventions that are in place at the beginning of the simulation, we define two sets of locally employed NPIs that lead to a reduction in contacts at home, school, work and other activities. These NPIs are implemented dynamically on a regional level. These sets of different strictnesses take effect when the number of currently infected individuals for a region exceed 35 and 100 per 100 000 individuals, respectively. This corresponds to an average contact reduction of 41% (range 34–47%) for threshold 35 and of 69% (range 63–75%) for threshold 100. For more details, also see the tables in the S1 Appendix. In addition, we consider the school holidays for each state, which are implemented similar to a school lockdown during this time. This means that all pupils as well as school staff will have reduced contact rates during this time.

**Probabilities for partially vaccinated and immunized populations.** The probabilities of partially and fully vaccinated persons to get exposed, asymptomatic, symptomatic, or, e.g., hospitalized are expressed as functions of these probabilities for unvaccinated persons and corresponding reduction factors $p$. In order to derive the conditional probabilities $\mu_{*_1{}_Y}^{*_2{}_Y}$ for $Y \in \{PV, V\}$ and partially and fully vaccinated individuals, we first define the probabilities

$$
\begin{aligned}
P(E_Y) &= p_{E_Y} P(E) \\
P(I_Y) &= p_{I_Y} P(I) \\
P(H_Y) &= p_{H_Y} P(H) \\
P(U_Y) &= p_{U_Y} P(U) \\
P(D_Y) &= p_{D_Y} P(D)
\end{aligned}
\tag{30}
$$

to get exposed, infected, hospitalized, critically infected, or to die after having already received the first or second vaccine dose. Here, $P(x)$, $x \in E, I, H, U, D$, is the *probability* for an unvaccinated and susceptible individual to end up in the respective compartment. Further, $p_x$, $x \in E_{PV}, I_{PV}, H_{PV}, U_{PV}, D_{PV}$, is a *reduction factor* for partially vaccinated persons to undergo a particular state and $P(x)$ is the resulting probability. For example, a parameter $p_{I_{PV}} = 0.8$ meant that a healthy individual who has received his or her first dose of vaccination is 20% less likely to get infected than an individual who has not received any vaccination yet.

Using conditional probabilities and elementary statistics, we obtain for both $Y \in \{PV, V\}$

$$
\rho_Y = p_{E_Y} \rho
\tag{31}
$$

as well as

$$
\begin{aligned}
\mu_{C_Y}^{I_Y} &= P(I_Y|C_Y) \overset{(*)}{=} P(I_Y|E_Y) = \frac{P(I_Y \cap E_Y)}{P(E_Y)} = \frac{P(I_Y)}{P(E_Y)} \overset{(30)}{=} \frac{p_{I_Y}P(I)}{p_{E_Y}P(E)} \\
&= \frac{p_{I_Y}}{p_{E_Y}}P(I|E) \overset{(*)}{=} \frac{p_{I_Y}}{p_{E_Y}}P(I|C) \overset{(**)}{=} \frac{p_{I_Y}}{p_{E_Y}}(1 - \mu_C^R) \\
\mu_{C_Y}^{R_Y} &= 1 - \mu_{C_Y}^{I_Y} \\
\mu_{I_Y}^{H_Y} &= P(H_Y|I_Y) = \frac{P(H_Y \cap I_Y)}{P(I_Y)} = \frac{P(H_Y)}{P(I_Y)} = \frac{p_H P(H)}{p_{I_Y}P(I)} = \frac{p_{H_Y}}{p_{I_Y}}P(H|I) = \frac{p_{H_Y}}{p_{I_Y}}\mu_I^H \\
\mu_{H_Y}^{U_Y} &= P(U_Y|H_Y) = \frac{P(U_Y \cap H_Y)}{P(H_Y)} = \frac{P(U_Y)}{P(H_Y)} = \frac{p_{U_Y}P(U)}{p_{H_Y}P(H)} = \frac{p_{U_Y}}{p_{H_Y}}P(U|H) = \frac{p_{U_Y}}{p_{H_Y}}\mu_H^U \\
\mu_{U_Y}^{D_Y} &= P(D_Y|U_Y) = \frac{P(D_Y \cap U_Y)}{P(U_Y)} = \frac{P(D_Y)}{P(U_Y)} = \frac{p_{D_Y}P(D)}{p_{U_Y}P(U)} = \frac{p_{D_Y}}{p_{U_Y}}P(D|U) = \frac{p_{D_Y}}{p_{U_Y}}\mu_U^D
\end{aligned}
\tag{32}
$$

Here, we used in $(*)$ that the probability to undergo state $C_Y$ or $C$, respectively, given $E_Y$ or $E$ is 1. In $(**)$, we used that only recovery, $R$, or symptom onset, $I$, are possible states from state $C$, which comprises pre- and asymptomatic cases.

The particular parameter ranges we use for the protection effects of partial and fully vaccination are based on three recent articles [26–28] and the systematic review [29]. The authors of [26] report a median effectiveness against symptomatic infection after one dose of vaccination between 31 and 48% for Alpha and Delta variant, respectively. We consequently vary $p_{I_{PV}} \in [0.3, 0.4]$. We assume a slightly reduced effectiveness for any infection of $p_{E_{PV}} \in [0.15, 0.25]$. From [28] and different values of effectiveness for AstraZeneca and Biontech vaccines, we take a weighted average and let $p_{H_{PV}} \in [0.85, 0.95]$. The recently published systematic review and meta analysis [29] reported pooled median effectiveness of fully vaccination of 66.9% against any infection, 75.5% against symptomatic infection and of 90.9% against hospitalization. We thus take $p_{E_V} \in [0.619, 0.719]$, $p_{I_V} \in [0.707, 0.807]$, and $p_{H_V} \in [0.859, 0.959]$. As the systematic review [29] reported that "No study reported admission to intensive care unit, intubation or death", we assume $p_{H_{PV}} = p_{U_{PV}} = p_{D_{PV}}$ and $p_{H_V} = p_{U_V} = p_{D_V}$.

In our model, people only die after admission to ICU. The case fatality rate (CFR) is therefore calculated by the chain of reduced probabilities of being hospitalized, then going to the ICU, and finally dying. Since certain parameters, such as the time span for critical courses of the disease, are by simplification assumed to be constant for unvaccinated and vaccinated individuals, we reuse certain parameters of Eqs (1)–(8) in Eqs (9)–(26) without introducing new variable names That means, that we use the time spans $T_{E_i}^{C_i}, T_{C_i}^{I_i}, T_{I_i}^{H_i}, T_{H_i}^{U_i}, T_{U_i}^{R_i}, T_{U_i}^{D_i}$ for the path of partially and fully vaccinated individuals; see also Fig 1.

Quantification of the infectiousness of vaccinated individuals is still actively studied [30–34]. While [30, 31] found evidence that viral load is similar in vaccinated and unvaccinated individuals, [32, 35] found substantially reduced viral loads in individuals that were vaccinated recently (about 2 weeks to 1–2 months ago). However, this effect was observed to decline with time distance to the vaccination event. On the other hand, [31, 33, 34] also found a faster decline of viral loads for vaccinated individuals, indicating a shortened time of high transmission potential. Summing up these findings, we consider a reduced time span for mild courses of the disease and (partially) vaccinated individuals. We thus reduce the time span for *carrier* or *infected* to directly recover after the transmission. In order to do so, we introduce the

parameter $\kappa$ = 0.5. To show the influence of this parameter, we will also present results for $\kappa$ = 1 which meant no change in infectiousness period. If not stated otherwise, in the results, we will use $\kappa$ = 0.5.

**Spatially resolved model.**   During the whole pandemic, infection dynamics were often highly heterogeneous even in single countries or federal states. As of, e.g., June 06, 2021, we saw a large number of German counties with incidences (weekly infections per 100 000 individuals) below and around 10 while about 5% of the counties showed infection dynamics with incidences of about 50 or higher; cf. [36]. As of November 04, 2021, incidences ranged from around 40 to around 700. In order to properly account for this fact, spatially resolved models have to be used.

To obtain a spatially resolved model, we define a graph where each node represents a single county in Germany and the edges represent the traveling and commuting activity from one county to another. Then, each single county will be attributed a full local model as described in Eqs (1)–(26). Commuter exchange and travel activities will be performed on a daily basis. According to the mobility from [37] and from geo-tagged tweets, we let a share of the population commute or travel to other counties. Severely or critically infected patients (state $H$ or $U$) are excluded from mobility. The approach was originally described in [6].

**Commuter testing**. We also briefly rephrase commuter testing as introduced in [15]. After having defined the share of the population to travel, we use a given probability to detect pre- or asymptomatic and symptomatic individuals. Using a generic mix of self-tests, individual and pool PCR-tests, we assume that 75% of the infected individuals are detected given a test of that day. The probability to detect infected individuals is then reduced according to the frequency of tests. For the scenario of one test per week, the probability to detect an infected commuter reduces to 15% per day. This share of detected individuals is then prevented from traveling and isolated in their home county. In practice, this is realized by reducing the $C_*$ and $I_*$ compartment and increasing the $C_*^+$ and $I_*^+$ compartment. For a visualization and more details, we refer to [15], in particular Fig 2. The compartment $C^+$ does not have any natural influx and only depends on the number of commuters and testing rates defined between counties on a daily basis. $I^+$ has only influx from $C^+$ and can also increase due to testing results.

**Numerical solution procedure of the local model.**   The system of nonlinear ordinary differential Eqs (1)–(26) is solved using the adaptive Runge-Kutta-Fehlberg45 (RKF45) method [38]. We use Monte Carlo runs with 500 simulations of sampled parameters from uniform distributions based on the ranges from Table 1.

**Vaccination process.**   As of November 04, 2021, full vaccination ratios in different federal states of Germany range from 57% to 78% [39]. The local vaccination rates used by us are based upon the officially reported numbers in [40]. The highly heterogeneous infection dynamics and also the largely varying vaccination ratios across different regions strongly advocate regionally resolved models. However, the data reported in [40] is not directly attributable to the German counties since vaccinations were reported with the county of vaccination and not by home location of the vaccinated; see [41]. While it can be expected that a large number of doses were administered to persons in their home county, a non-negligible number of doses were administered at work places or, e.g., in neighboring counties. However, it is difficult to make precise assumptions on the true number of vaccinations per county.

Since country- or federal state-wide averages may be averaging out some local effects, we decided to use intermediate-sized region averages. For this, we use a set of labor market regions [42, 43] which are built such that interactions within the regions are strong and connections with other outside regions are few. We have taken the 34 regions as provided by [43]. Due to an unrealistically high number of reported vaccinations (i.e., more vaccinations than inhabitants) in the labour market region of Ulm, we aggregated the regions of Ulm and

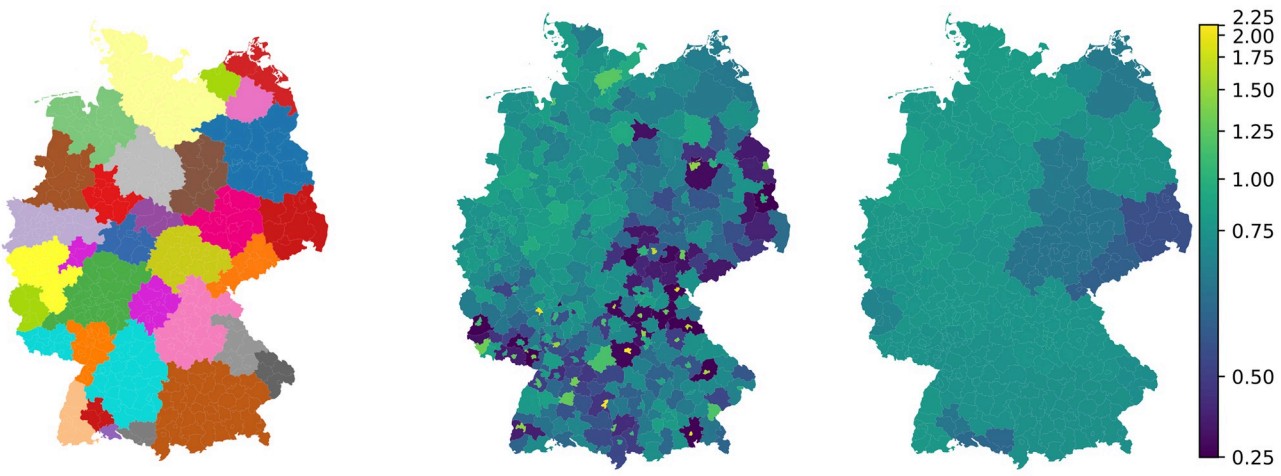

**Fig 2. Intermediate-sized labor market regions (left; cf. [43]).** Reported (center) and approximated (right) vaccination ratios for age group 18–59 as of November 02, 2021. Ratios computed by dividing vaccinations up to November 02, 2021 by local population size. Using geodata "Verwaltungsgebiete 1:2 500 000, Stand 01.01. (VG2500)" from https://gdz.bkg.bund.de, copyright; GeoBasis-DE / BKG 2021, license dl-de/by-2–0, see https://www.govdata.de/dl-de/by-2-0.

Stuttgart. The remaining 33 intermediate-sized regions are depicted in Fig 2 (left). Over these regions, we have computed age-dependent averages of full vaccination (in general, this means second-dose vaccination) for every day since the beginning of the vaccination process in Germany. The averaged vaccination ratios for age group 18–59 as of November 02, 2021 are depicted in Fig 2 (right).

The population data we use here and in our simulations is the official extrapolation of age-resolved county population in Germany [44].

In order to not overparametrize the model, we only consider one vaccination path, i.e., one generic vaccine. As an average from about 80% of administered mRNA and 20% of administered vector vaccines [45], we take an average interval $T_{PV}^V$ between first and second dose of 7 weeks (49 days).

For near-future scenarios, we compute the age-dependent vaccination rates based on the vaccination rates of the previous weeks. This is a reasonable assumption given the slow but rather constant vaccination speed of the last months [45].

The vaccination process in our model happens on a daily basis and not in the ordinary differential equations. In the previous steps, we have explained how to obtain daily vaccination numbers for the different counties and age groups. Let us consider an arbitrary but fixed age group and county and let $\mathcal{D}_2(t)$ be the number of full vaccinations on day $t$. The number of people $\mathcal{D}_1(t)$ that receive their first vaccine dose on day $t$ is obtained from

$$\mathcal{D}_1(t) = \mathcal{D}_2(t + T_{PV}^V). \tag{33}$$

Similar to the commuting step, we adapt the subpopulations in a daily vaccination step by

$$
\begin{aligned}
S(t) &\leftarrow S(t) - \mathcal{D}_1(t), \\
S_{PV}(t) &\leftarrow S_{PV}(t) + \mathcal{D}_1(t) - \mathcal{D}_2(t), \\
S_V(t) &\leftarrow S_V(t) + \mathcal{D}_2(t).
\end{aligned}
\tag{34}
$$

Note that age group 0–4 years is excluded from vaccination for now and from age group 5–14 years only children above 12 years can receive vaccination from about August 2021 on [46].

## Alpha and Delta variant

To account for the Alpha variant in Germany [47, Report of Apr. 7], we use a 1.4 times increased value for the transmission risk $\rho_i$ [48] compared to the wild-type considered in [6]. The Delta variant of SARS-CoV-2 has already reached Germany and made up approximately 48%–59% of the different variants by July 4 [49, Situation Report of July, 7]. Different studies tried to assess the increased infectiousness of the Delta variant. The authors of [8] found a 55% increased reproduction number of Delta compared to Alpha, while even observed 58–120% increased effective reproduction numbers and [9] agreed on a range of 40–60% and [7]. In our simulations, we will therefore consider Delta to be 40–60% more infectious than the previously dominant Alpha variant. In the retrospective analysis, it was safe to assume that Delta continued to increase its share exponentially as it did in the UK [50]. We assume that its share $\sigma_\delta$ doubles each week, so

$$\sigma_\delta = \min\left(1, \frac{2^{\frac{t}{7}}}{100}\right) \tag{35}$$

is the share of the Delta variant from June 06, 2021, on. The relative transmission risk of the Delta variant compared to the Alpha variant is denoted $v_\delta$.

Our model reflects this development by increasing the transmission risk $\rho_i = \rho_i(t)$ at each day $t$. Including the seasonality $s_k(t)$ by (27) and (28), we obtain

$$\rho_i(t) = (1 - \sigma_\delta)\, s_k(t)\, \rho_i^{(0)} + \sigma_\delta\, s_k(t)\, v_\delta\, \rho_i^{(0)}, \tag{36}$$

where $\rho_i^{(0)}$ is the base transmission risk of the Alpha variant for age group $i$ as given in Table 1.

## Results

In the following, we present different simulations of our extended model that includes the possibility of infection after full vaccination. Simulations for 90 days from June 06, 2021, onwards were already available in the first version of this paper submitted in July in which we allowed for infections after partial vaccination only. From today's point of view, these are retrospective scenarios. However, compared to the original results, we only extended the first version model by possible (re)infection after immunization and adapted the vaccination process and vaccine efficacy to the most recent study results.

Also, we have corrected the expected vaccination progress by the real progress observed. Unlike many other European countries, the vaccination progress in Germany slowed down considerably shortly after the first version of this paper was submitted. While we had seen 10.1m first vaccinations between June 08, 2021, and July 09, 2021, there were only 3.4m further first vaccinations until August 09, 2021 [45].

For all curves, if we present *Infected* or *ICU* information, we aggregate the results for age group $i$ over the compartments $I_i$, $I_{PV,i}$ and $I_{V,i}$ and $U_i$, $U_{PV,i}$, $U_{V,i}$, respectively.

The number of individuals in the different compartments at the start of the simulations are extrapolated from the RKI [51] and DIVI [52] database. Please note that RKI and DIVI only report case numbers of positive tested individuals and individuals admitted to ICU. Our

extrapolation formulas are rather intuitive by shifting time series and applying the probabilities to undergo states that are not reported in the input sources (e.g. *exposed*). For a detailed explanation of the initialization process, see the S2 Appendix. The full code for extrapolation is freely available in [53].

Our aim is to analyze the effect of different relaxation strategies of NPIs. Therefore, we define the following scenario parameters:

1. Timing of the lifting of locally employed NPIs: Regional, dynamic NPIs are no longer decreed either from July 01 or August 01 on. Lifting these means that the contact patterns return to prepandemic contact patterns.

2. Testing commuters: We either test individuals coming from counties where local NPIs are in place once a week or we do not test commuters at all. Commuters will be isolated if tested positive.

3. Wearing masks: We consider the two cases where masks are continued to be worn after the local NPIs are lifted or not at all.

The factor space of the above items has a dimension of eight. In the following, we focus on four out of these eight scenarios, where we combine masks and testing, since their impact proves not to be significant enough on their own. For each scenario, we do 500 Monte Carlo runs.

We assume that Delta will make up more then 50% of the different variants within 40 days on July 16 and over 80% just 4 days later on July 20. In England, infection numbers began to rise again from day to day when Delta made up 80%. We expect to see a similar effect in our simulations. If all NPIs are lifted too soon, we expect that the new Delta variant will lead to a significant increase in the number of infections despite the ongoing vaccination process and especially in the younger age groups. Even the summer school holidays during the simulation period might not be enough to counteract the much more infectious variant.

We will first present each scenario for the Delta variant to be either 40% or 60% more infectious than the Alpha variant; see Figs 4, 5, 7, and 8. Here, we will also consider the difference between (partially) vaccinated individuals having shorter periods of infectiousness; see Figs 5 and 6. Then, we will present age-resolved cumulative numbers of infections for the different scenarios in Fig 9. Finally, we will present three different scenarios about possible future development of the current wave of infections assuming Delta to be 60% more infectious than the Alpha variant.

As a basis for further analysis, we first consider Fig 3, where we depict the number of people who have received their first dose of the vaccine, the people who are fully vaccinated to the virus, and the number of pupils on vacation in Germany. The number of only once vaccinated

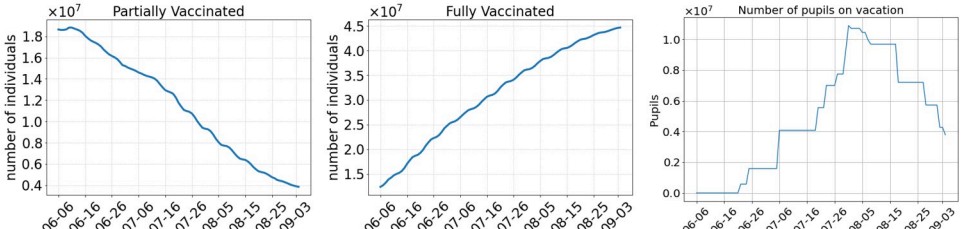

**Fig 3.** Left: Number of individuals that have received the first dose of the vaccine at least two weeks ago and who are not yet fully vaccinated. Center: Number of individuals that have received their second dose of the vaccine at least seven days ago. Right: Number of pupils that are currently on summer vacation.

individuals $\mathcal{D}_1(t)$ in Eq (33) constantly drops from about 20 million beginning of June to about 400 thousands beginning of September. On the other hand, we have about 45 million fully vaccinated individuals in the beginning of September. Since all federal states in Germany have individual dates for their summer holidays, we provide the number of pupils on holiday on the right of Fig 3. From 16 June onward more and more pupils will be on holiday with the maximum reached by the end of July.

Please note that the following evaluation is qualitative and essentially based on the median results. With a simulation of 90 days, uncertainties accumulate. Many other scenarios are possible as the p25 and p75 percentiles indicate. Note, however, that we sample all contact reduction factors from intervals. As contact reduction is one of the most important factors in mitigation, the median may correspond best to the contact reduction in the center of the given intervals while p75 and p25 percentiles result from higher or lower sampled contact reduction. As it is difficult to present all subtle details in a reasonably limited number of images, we also provide additional information on the median results in the text.

**Scenario 1**. We start analyzing the scenario with the weakest set of NPIs, namely no commuter testing before and a lifting of the regional NPIs in the beginning of July, without keeping masks after opening (Scenario 1). This basically equates to returning to prepandemic patterns. The number of infected, ICU admissions and deaths during the simulation are depicted in Fig 4 (top) for an assumed 40% more infectious Delta variant. The median run indicates that the early opening leads to a rise of infections from about 6–16 July onward. This leads to a continuous increase in the number of deaths. ICU admissions first decrease but start to rise as well by the end of July. If we assume that Delta is 60% more infectious (Fig 4, bottom), by the end of the simulation, there are about two or three times as many infected people as if we assume a 40% higher infectiousness of Delta.

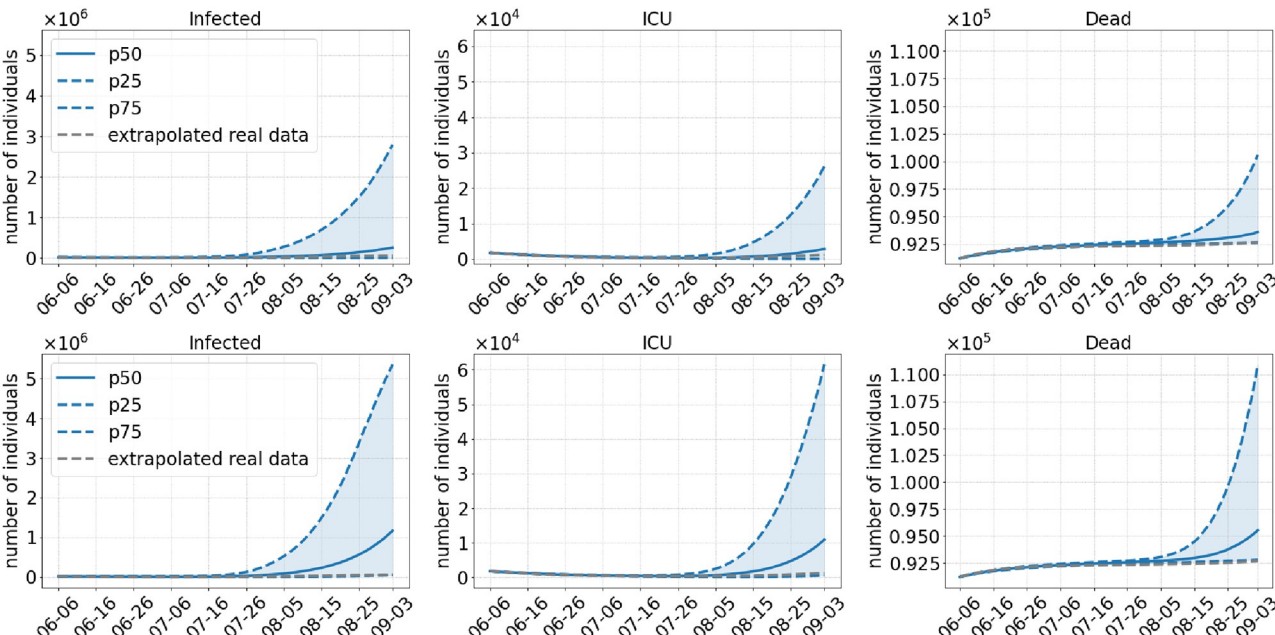

**Fig 4. Results of Scenario 1: No commuter testing, local NPIs decreed until July 01, 2021 only, no masks after opening.** The median of 500 runs is shown in solid blue; the blue dotted lines are the 25% and 75% percentiles. The top row shows results with an assumed 40% more transmissible Delta variant compared to 60% in the bottom row.

**Scenario 2**. As Scenario 1, Scenario 2 also dismisses all regional NPIs in July. In contrast to Scenario 1, commuter testing will be done with local NPIs before and masks will continue to be worn after opening. The prior testing and the continuation of mask wearing already lead to a substantial decrease in infection numbers by the end of the simulation; cf. Fig 5. While this scenario may be closest to reality from a retrospective view, simulated average infection numbers slightly underestimate the real development. However, there are two effects that have not yet been modeled and which can probably explain the slight differences in the curves. As [41] reports for the summer period, a non negligible number of infections had been imported from other countries. Between 17% (beginning of July) and 24% (end of August) of the detected cases were traced back to foreign countries. This effect is difficult to model since reliable infection and tourist numbers for all holiday destinations had to be determined. The corresponding intra-Germany infection dynamics should then slightly underestimate the true dynamics. Furthermore, an increased testing of pupils after the end of the holidays may have lead to a temporary break of infection rise. For ICU and death numbers, we find a good prediction for the first weeks of the simulation period, with a slight underestimation of ICU numbers and a slight overestimation for death numbers. However, the qualitative predictions for after three months are quite reasonable.

With Figs 5 and 6, we show the different development of infections numbers if vaccinated individuals were infectious as long as unvaccinated ($\kappa = 1$). In the previous sections and based on [30–34], we laid out that $\kappa < 1$ is a reasonable assumption. In Fig 6, we see how the same infectious period for vaccinated and unvaccinated would have lead to a steep increase in infections already in summer.

**Scenario 3**. Scenario 3 postpones the relaxation of all regional NPIs to August and, similar to Scenario 1, does not do any commuter testing and lifts wearing of masks after full opening in August. The median results of Fig 7 (top) show a clear downward trend in case numbers as well as in ICU admissions. The 75% percentile shows a slow rise but only up to the detected cases numbers and ICU admissions. In the median, this also holds for Delta-60% (Fig 7, bottom) up to mid of August. Then, however, infection numbers show a steep increase. The increase is slow at first and gets steeper by August 15 when the number of pupils on vacation begins to drop. Again, this leads to an increase in the number of ICU admissions. We see that Scenario 3 would have kept the infection numbers much smaller until mid of August, but then, the discontinuation of masks would have led quickly to degenerated virus dynamics over the month of September.

**Scenario 4**. In Scenario 4, we analyze the most strict set of NPIs. In addition to a late opening in August we test commuters as long as local NPIs are decreed and we even keep the mask mandate until the end of the simulation. As expected, this scenario has the greatest chance of continuously preventing another wave of infections. Fig 8 on the bottom shows that even in the worst case, predicted by the 75%-percentile with Delta-60%, the number of infections at the end of the simulation are close to the number of the detected cases. A similar argument can be given for the ICU admissions.

**Scenarios 1, 2, 3, and 4**. In Fig 9, we clearly see that school children are hit the hardest even if schools were closed for holidays for up to six weeks during the simulation. From the comparison of the results for Delta-40% (left) and Delta-60% (right), we see the huge difference that is made by a 60% more transmissible virus compared to 40%. Comparing Scenario 1 and 2, we see that the continuation of wearing masks and keeping distance has an important effect. The median results of Scenario 3 and 4 seem to be quite close, but 75% percentiles as well as the steep increase in case numbers from mid of August for the Delta-60% case (cf. Fig 7) indicate a new wave of infections after the simulation period.

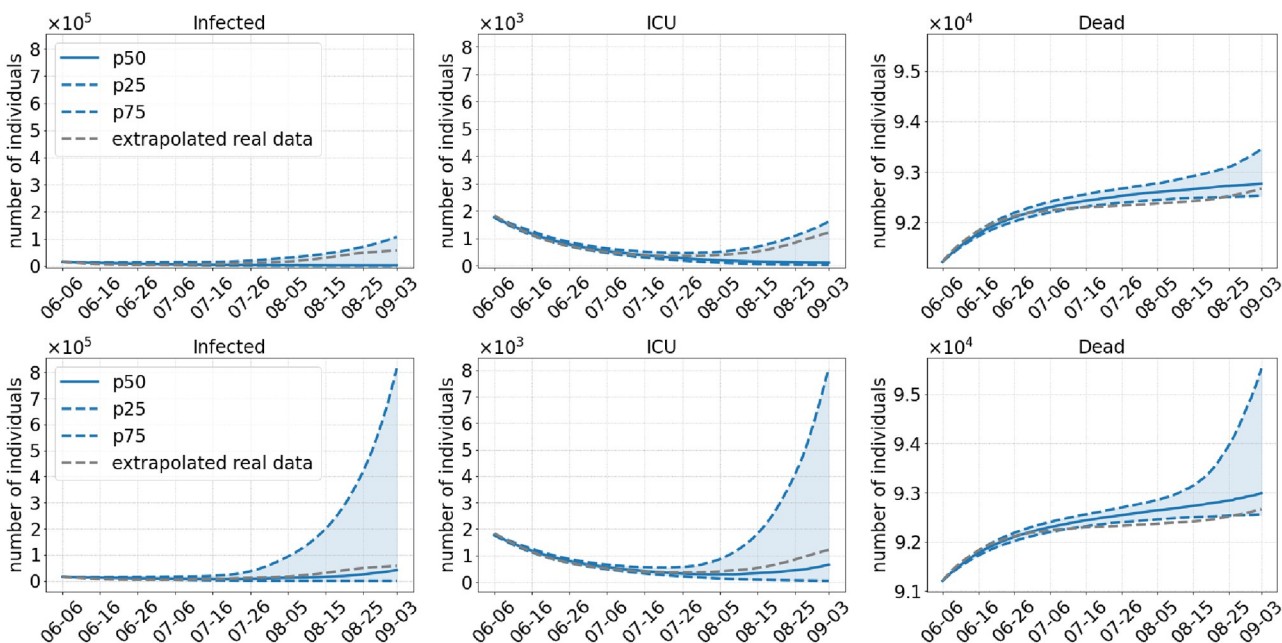

**Fig 5. Results of Scenario 2: Commuter testing once a week with local NPIs decreed until July 01, 2021, keeping masks and distancing after opening.** The median of 500 runs is shown in solid blue; the blue dotted lines are the 25% and 75% percentiles. The top row shows results with an assumed 40% more transmissible Delta variant compared to 60% in the bottom row.

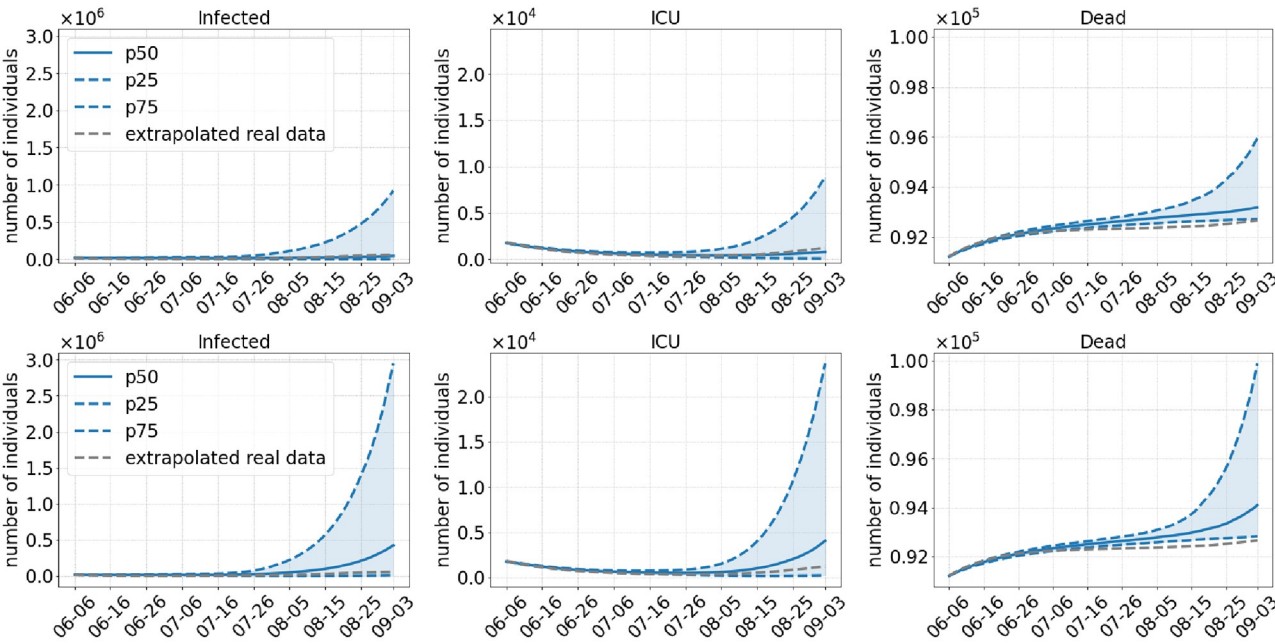

**Fig 6. Results of Scenario 2 with identical time span for carrier and infected individuals either unvaccinated or partially or fully vaccinated; $\kappa = 1$; cf. Fig 1: Commuter testing once a week, local NPIs decreed until July 01, 2021, keeping masks and distancing after opening.** The median of 500 runs is shown in solid blue; the blue dotted lines are the 25% and 75% percentiles. The top row shows results with an assumed 40% more transmissible Delta variant compared to 60% in the bottom row.

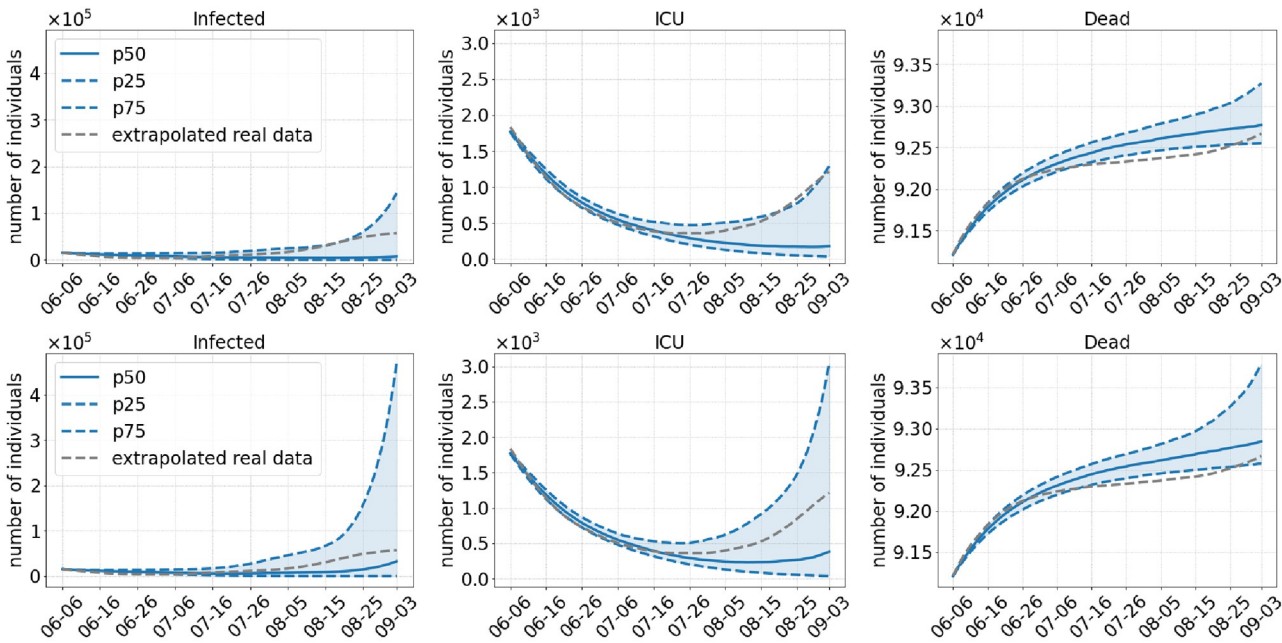

**Fig 7. Results of Scenario 3: No commuter testing, local NPIs decreed until August 01, 2021, no masks after opening.** The median of 500 runs is shown in solid blue; the blue dotted lines are the 25% and 75% percentiles. The top row shows results with an assumed 40% more transmissible Delta variant compared to 60% in the bottom row.

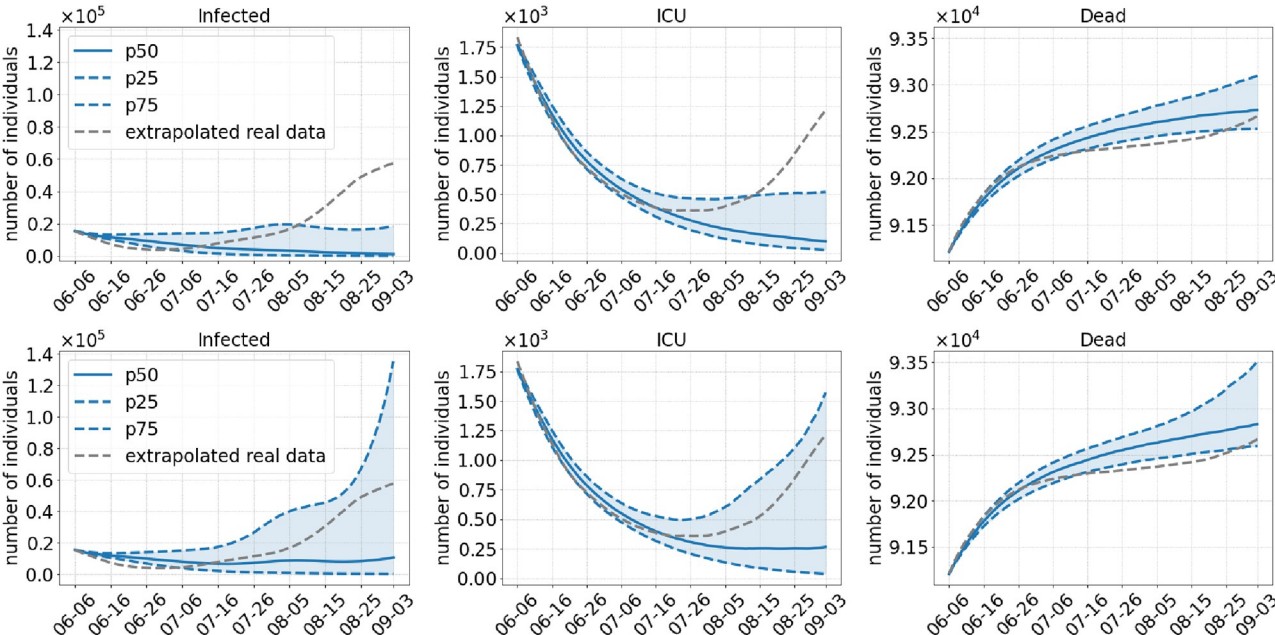

**Fig 8. Results of Scenario 4: Commuter testing once a week with local NPIs decreed until August 01, 2021, keeping masks and distancing after opening.** The median of 500 runs is shown in solid blue; the blue dotted lines are the 25% and 75% percentiles. The top row shows results with an assumed 40% more transmissible Delta variant compared to 60% in the bottom row.

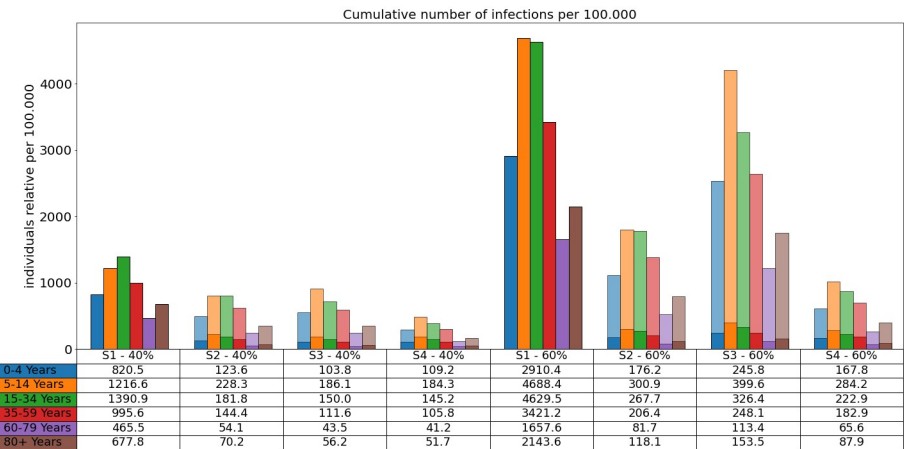

| | S1 - 40% | S2 - 40% | S3 - 40% | S4 - 40% | S1 - 60% | S2 - 60% | S3 - 60% | S4 - 60% |
|---|---|---|---|---|---|---|---|---|
| 0-4 Years | 820.5 | 123.6 | 103.8 | 109.2 | 2910.4 | 176.2 | 245.8 | 167.8 |
| 5-14 Years | 1216.6 | 228.3 | 186.1 | 184.3 | 4688.4 | 300.9 | 399.6 | 284.2 |
| 15-34 Years | 1390.9 | 181.8 | 150.0 | 145.2 | 4629.5 | 267.7 | 326.4 | 222.9 |
| 35-59 Years | 995.6 | 144.4 | 111.6 | 105.8 | 3421.2 | 206.4 | 248.1 | 182.9 |
| 60-79 Years | 465.5 | 54.1 | 43.5 | 41.2 | 1657.6 | 81.7 | 113.4 | 65.6 |
| 80+ Years | 677.8 | 70.2 | 56.2 | 51.7 | 2143.6 | 118.1 | 153.5 | 87.9 |

**Fig 9. Total number of infections in the different age groups over the duration of the simulation.** Bar plots: Number of infections per 100.000 individuals over 90 days and each scenario for Delta 40% (left) and 60% (right). The transparent bars are the 75% percentiles (for the sake of scaling omitted for Scenario 1) and the solid bars are the median values. Table: The median values are also presented in the table below the plot.

In all scenarios, we overestimate the number of infected individuals of the age group 80+. Age-resolved vaccination data are only stratified in age groups 12–17, 18–59 and 60+. In the consequence, we cannot assess the vaccination ratios in the distinguished age groups 60–79 and 80+. Also, the studies on the effectiveness of the vaccine are not age-resolved, which would be needed to better fit our model.

Comparing the curves for the number of deaths, our simulated median slightly overestimates the extrapolated real data. For Scenario 2, we predict between 1554 and 1801 new deaths while the extrapolated real data shows 1455 deaths. However, the source data does not provide the exact date of deaths which we have to extrapolated from the day of assumed infection. Also, deaths may be reported with a substantial time delay. We already discussed this issue in [6]. Comparing the number of reported deaths in the daily situation reports of June 06 and September 03, 2021, we even get 3079 reported deaths for the period considered [36, 54].

**Scenarios 1F, 2F, and 3F**. We now consider three different model scenarios S1F, S2F, and S3F for the future development. We have started our simulations on October 15, 2021, and compute results up to January 12, 2021; see Fig 10. In all scenarios, masks and distancing are applied in *school*, *work*, and *other* (everything except home, school and work) locations but not in *homes*. For the Scenario 1F, an additional contact reduction between 5 and 15% is assumed in *school*, *work*, and *other*. For the Scenario 2F, contact reduction in *schools* and *other* is increased to 35–45% while home office is kept at an average of 10%. For the Scenario 3F, only home office is increased from 5–15 to 35–45% (compared to the Scenario 2F).

In the first Scenario 1F, we continue to wear masks and keep distance but only reduce contacts by about 10% in most locations. Here, almost all school children will have been infected by the end of the simulation and a large majority of most other age groups, too. Reducing contacts in *schools* and *other* locations by 40% on average, can already substantially reduce the number of infections (Scenario 2F).

Most interesting is the effect of Scenario 3F. Keeping the settings of Scenario 2F, while only increasing the share of home office from 10% to 40% on average, reduces the number of infections in the working population by almost a third. We refer to the analysis of [55] from which a home office potential of at least 40% of the population could be assumed. Although school

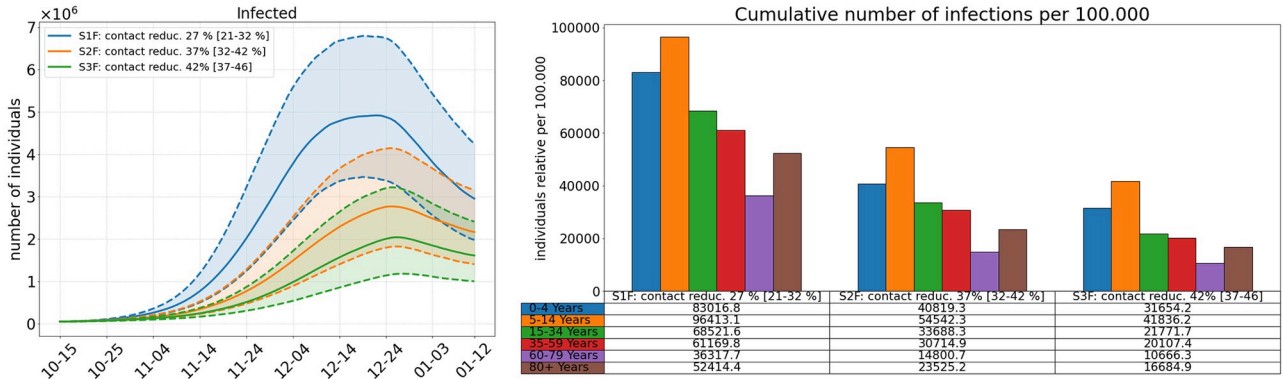

**Fig 10. Comparison of three different model scenarios for future developments assuming Delta to be 60% more infectious than the Alpha variant.** In all scenarios, masks and distancing are applied in *school*, *work*, and *other* (everything except home, school and work) locations but not in *homes*. For the Scenario 1F, an additional contact reduction between 5 and 15% is assumed in *school*, *work*, and *other*. For the Scenario 2F, contact reduction in *schools* and *other* is increased to 35–45% while home office is kept at an average of 10%. For the Scenario 3F, only home office is increased from 5–15 to 35–45% (compared to the Scenario 2F). **Left**: Median simulation results in solid lines for the number of infected individuals over the period October, 15, 2021 to January 12, 2021. Percentiles p25 and p75 shown in dashed lines. **Right**: Total number of infections per age group over the whole simulation period. The median values are also presented in the table below the plot.

children are not directly affected by that NPI, we even see a 24% reduction in case numbers of school children aged 5–14.

## Discussion

With ongoing vaccination in Germany counteracted by the possible spread of new SARS-CoV-2 variants, we are faced with the decision on when regional NPIs can be safely lifted or NPIs like face masks and testing can be relaxed. This question is investigated in our paper, and we provide a qualitative answer by comparing different scenarios under the assumption that the Delta variant will be dominant in a few weeks from June 2021 on. In the revision, we extended the model to allow for reinfections and infections after full vaccination. We further included three model scenarios with different contact reduction factors to see the potential impact of the current wave of SARS-CoV-2.

Our results show that an opening with the removal of all NPIs including masks would have been too early in July and would have lead to another wave of infections (Scenario 1). Due to the high discrepancy between the confidence intervals in Fig 4, it is hard to predict the magnitude of this new wave. However, we can safely assume that the number of infections would have grown even faster after the end of the simulation by the end of August as these results are damped by the summer school holidays between July and September. Summer holidays were particularly effective since pupils below 12 were not yet vaccinated in Germany and pupils aged 12 and older only started to get vaccination at the end of our simulations. For all scenarios the incidences are highest or second to highest (only S1—40%) in the age group 5–14, although this age group benefits the most from the school holidays. With our extended model that allows for infections after recovery or full vaccination, we also see increases in ICU and death numbers, the more infection numbers rise.

It should be noted, however, that even with a lower death toll, there may be significant costs to public health by long-term consequences such as post-covid syndrome (PCS) that prevents approximately 11% of non-hospitalized patients from returning to work more than half a year after their infection [56]. Additional measures like wearing masks and testing commuters after the opening in July (Scenario 2) help to reduce the number of cases substantially.

Postponing the relaxation of regional NPIs to the first of August also has a great benefit to the reduction of case numbers. By the first of July, only about 20–25 million people were fully vaccinated in our scenarios, whereas there are about 35 million by the first of August as shown in Fig 3. In addition, the overall transmission of the virus is reduced even further with the late opening, since the two age groups that are over 60 years old and that have a higher risk of getting infected then people from younger age groups [57] will be vaccinated with even higher ratios. However, if we dismiss wearing masks and testing immediately (Scenario 3) and if we assume that Delta is 60% more infectious, we might still see a rise in the number of infections when Delta is accountable for 80% of the infections on 20 July. The safest scenario in our simulations is the fourth one where we open by the first of August but continue to wear masks and test commuters. However, slower but new infection waves cannot be excluded.

Even though the difference in the median of Scenarios 2 through 4 is fairly small, there is still a real world risk of another wave of infections based on the increase in case numbers of the higher percentiles.

In all of our scenarios we see a clear shift of infections from the older to the younger age groups and especially to the school children. The quantity of this shift depends on Delta's infectiousness and NPIs but in the worst cases the age group 5–14 is overrun by an infection wave.

Our modeling approach aims at providing a data-based comparison of different scenarios for lifting regional NPIs with or without the continuation of wearing masks. With the extended model, we now allow for infections and severe courses after recovery or full vaccination.

Given the current situation, we provided three model scenarios for future developments. All of these scenarios are insufficient in breaking the current wave of infections, even though the third one already reduces contacts by 42% on average. What we can infer from these scenarios is, first, that school children can be protected by a substantial increase of remote working individuals. A current limitation of our predictions is that we do not yet consider the effect of mandatory testing at workplaces. Second, for breaking the current wave of infections, we either need contact reductions which are substantially higher than 42% or a substantial increase in vaccinations and tests. For an analysis on test frequency, see also [15].

A limitation of our models and parameters is that we do not have reliable data on the age-resolved vaccine effectiveness and that we do not yet include reduced vaccine effectiveness with distance to the vaccination event nor booster or third vaccinations. Furthermore, we do not model testing of pupils or general individuals. In the scenarios for the summer holidays, the testing of pupils was less relevant. We do neither consider border regions, i.e., the impact of systematically higher incidences in a neighbouring country. However, this effect may be secondary. On the one hand, incidences are now often higher in Germany than in neighboring countries. On the other hand, data from official sources [37] report the number of incommuters from foreign countries to certain border-near regions to be about 10% of the total number of incommuters. For the summer period in particular, we cannot account for all travelling activities during the holidays.

Another limitation of our work is the exact quantification of the strength of distinct NPIs. While the timing at which the NPIs are lifted is fairly straightforward and testing can be achieved by isolating a portion of the infected commuting population, other NPI related effects e.g. wearing masks or social distancing are achieved by reducing the number of contacts. The effect of wearing masks corresponds to a contact reduction in schools, at work or at other locations which was sampled from a uniform distribution between 25%–35% in our simulation and can therefore be replaced by other NPIs that yield the same amount of contact reduction.

We do not aim to predict exact infection numbers, but we provide a comparative evaluation of how the timing of NPI dismissals increase or reduce the likelihood of a further spread

SARS-CoV-2 in the light of the Delta variant and vaccination. For the future scenarios, we show the effect of different contact reduction strategies which must be complemented by increased vaccination and testing numbers. In all scenarios, it is the school children who have not yet been vaccinated in whom a fourth wave triggers the highest numbers of infections.

## Conclusion

In this study, we analyzed different strategies for lifting and reintroduction of of nonpharmaceutical interventions that were in place during the SARS-CoV-2 pandemic, while accounting for the new Delta variant and the ongoing vaccination process. We have shown that at the current rate of vaccination, there is still a great risk of further waves of infections if NPIs are lifted too early. The relation of deaths to infections will be reduced compared to previous waves due to the vaccination process. Nevertheless, with Delta having taken over, it seemed advisable to keep wearing masks and keeping distance for some further time after lifting all other restrictions in summer to ensure the population's safety. Given the current rise of infections in winter, we even face the reintroduction of new NPIs. Due to the many uncertainties regarding the simulated results, e.g., the true risk of infection of the new Delta variant, the seasonality or even the compliance of the population, it is of paramount importance that we continue to monitor the real-world dynamic of the pandemic, continue the vaccination process as fast as possible and adopt the necessary NPIs accordingly.

Before autumn, it appeared appropriate to take preventive hygiene measures in preparation of school openings in order to allow for a sustainable education. We have missed the chance to obtain a sufficient level of vaccinated individuals before autumn. As in summer, we need to protect the health of school children and their right to normal school operation. In all our scenarios rising infection numbers will continue to hit school children the hardest and the seasonality will further drive the infections.

From the scenarios computed for the future development of the SARS-CoV-2 spread, we infer that school children can be protected partially by a substantial increase of remote working individuals. However, a simple contact reduction as of 42% on average will not be sufficient to break the current wave of infections. We either need contact reductions which are even higher than 42% or a substantial increase in vaccinations and tests.

## Supporting information

**S1 Appendix. NPI parameters for the different scenarios and time spans.**
(PDF)

**S2 Appendix. Initialization of compartment models from confirmed cases, deaths, ICU and vaccination numbers.**
(PDF)

## Author Contributions

**Conceptualization:** Wadim Koslow, Martin J. Kühn, Sebastian Binder, Margrit Klitz, Achim Basermann, Michael Meyer-Hermann.

**Data curation:** Wadim Koslow, Martin J. Kühn, Sebastian Binder, Daniel Abele.

**Formal analysis:** Wadim Koslow, Martin J. Kühn, Daniel Abele.

**Funding acquisition:** Martin J. Kühn, Sebastian Binder, Margrit Klitz, Michael Meyer-Hermann.

**Investigation:** Wadim Koslow, Martin J. Kühn, Margrit Klitz.

**Methodology:** Wadim Koslow, Martin J. Kühn, Sebastian Binder, Margrit Klitz, Daniel Abele, Michael Meyer-Hermann.

**Project administration:** Martin J. Kühn, Margrit Klitz, Achim Basermann, Michael Meyer-Hermann.

**Resources:** Margrit Klitz, Achim Basermann.

**Software:** Wadim Koslow, Martin J. Kühn, Daniel Abele.

**Supervision:** Martin J. Kühn, Sebastian Binder, Margrit Klitz, Achim Basermann, Michael Meyer-Hermann.

**Validation:** Wadim Koslow, Martin J. Kühn, Sebastian Binder, Margrit Klitz, Daniel Abele, Michael Meyer-Hermann.

**Visualization:** Wadim Koslow.

**Writing – original draft:** Wadim Koslow, Martin J. Kühn, Sebastian Binder, Margrit Klitz.

**Writing – review & editing:** Wadim Koslow, Martin J. Kühn, Sebastian Binder, Margrit Klitz, Daniel Abele, Achim Basermann, Michael Meyer-Hermann.

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
