## [Decision Letter · Decision Letter 0]

17 Sep 2021

Dear Mr. Koslow,

Thank you very much for submitting your manuscript "Appropriate relaxation of non-pharmaceutical interventions minimizes the risk of a resurgence in SARS-CoV-2 infections in spite of the Delta variant" for consideration at PLOS Computational Biology.

As with all papers reviewed by the journal, your manuscript was reviewed by members of the editorial board and by several independent reviewers. In light of the reviews (below this email), we would like to invite the resubmission of a significantly-revised version that takes into account the reviewers' comments.

We cannot make any decision about publication until we have seen the revised manuscript and your response to the reviewers' comments. Your revised manuscript is also likely to be sent to reviewers for further evaluation.

Sincerely,

Claudio José Struchiner, M.D., Sc.D.

Associate Editor

PLOS Computational Biology

Virginia Pitzer

Deputy Editor-in-Chief

PLOS Computational Biology

Reviewer's Responses to Questions

**Comments to the Authors:**

Reviewer #1: Dear Authors,

I have reviewed the manuscript assigned. In this manuscript, the authors aim to evaluate the burden of COVID19 as NPIs are lifted in Germany. While the topic of the study is not novel (i.e. evaluating the lifting of NPIs in view of increasing vaccination coverage), the manuscript is specific to Germany with a country-specific model parameterization. The model itself is highly detailed taking into account demographics, vaccination, seasonality, and spread of disease between geographic regions within Germany. The model is well-parametrized with plausible values and relevant citations.

I have a few minor comments for the authors, particularly in the model formulation and implementation. I would like to mention that I was not given access to an appendix, if one is available. Perhaps, some of my comments are already addressed in the appendix.

-- It is not clear to me how the Spv compartment changes as individuals are vaccinated. There should be an interaction where a certain number of individuals from the S class move to the Spv class on a daily basis. Indeed, the authors state "a portion of the susceptible population S is transferred to the partially vaccinated compartment Spv" (presumably, the rate here is at the daily or weekly resolution). However, the model equations (Eq 9) does not seem to model this dynamic. Infact, it only has a negative term, suggesting the compartment always decreases from the initial value. Probably a constant term needs to be added to the equation (or an interaction term between the S and Spv compartments).

-- Similarly, other model dynamics are not clear from the equations. For example, how do the compartments C+, I+ interact in the model? Where is the contact matrix in the equations? Perhaps the authors can expand their methodology and include more details in an appendix. Moreover, I understand the current model is an extension of a previously published model. However, for the sake of clarity, the model should be described fully again instead of referring back to another paper.

-- On a similar note, it would be nice to also have the table of all the parameters used in the model without switching/searching for other papers.

-- The authors state that parameters, when applicable, were sample from Uniform Distributions. I am curious to see how the results would change if the authors used more specific distributions for disease-specific parameters (for example, Lognormal or Gamma for the incubation period). Almost all parameters pertaining to COVID19 have been fairly described at this point.

-- In Scenario 3, the authors assume that "wearing masks equates to a 20% to 40% reduction in contacts in the categories school, work and other". First, is there a relevant citation for this? Second, how was this implemented? Was the effectiveness sampled randomly from a Uniform Distribution for each MC simulation?

-- Is it necessary for public-health units in Germany to see the temporal trajectories of in Fig 4? May I suggest plotting, in a single figure, the cumulative number of infections for each age-group... and providing a temporal trajectory for all the age groups summed together.

-- Please upload the relevant codes and libraries required for reproducibility of the manuscript. Perhaps these files could be hosted on an online repository service such as GitHub.

Reviewer #2: PCOMPBIOL-D-21-01275: Appropriate relaxation of non-pharmaceutical interventions minimizes the risk of a resurgence in SARS-CoV-2 infections in spite of the Delta variant

In this study the authors analyse the transmission dynamics of COVID-19 in Germany under NPI relaxation, an increasing number of vaccinated individuals and in the presence of Delta variant. After some simplifying assumptions, the authors conclude that there is still a great risk of another wave of infections if NPIs are lifted too early. The current version of the manuscript has major methodological issues that must be addressed as depicted below.

It is clear from many reports that the efficacy of all vaccines are lower than 100% independent of the variant. Additionally, there are some evidences that vaccines induce more protection than naturally acquired infections (10.1126/scitranslmed.abi9915). The model design assumes 100% immunity for both naturally infected or vaccinated individuals. Although the authors state that they do not intend to provide precise forecasts, these assumptions might have a significant impact on model inferences and conclusions. Moreover, it seems to me that there are no methodological limitations which prevent the authors to account for more realistic scenarios where no full immunity is acquired after infection/vaccination. Please, clarify it.

The authors assume that the vaccination capacity remains constant since the beginning of the simulations. This is a plausible assumption. On the other hand, although not stated in the text, the authors also implicitly assume that the number of individuals getting vaccinated remains constant over time. Like other countries, vaccination rates have substantially decreased in Germany after reaching 60% coverage and this fact might also impact model inferences. Please, clarify it and acknowledge limitations if needed.

Regarding the system of equations present on pages 4 and 5:

1. In equations 1 and 9, the term (Cj + Cpv,j) is multiplied by 0.5. This value is not described in the text and, apparently, it is missing in equation 2 and 10. Please, clarify it.

2. Equation 1 as presented does not account for (i) individuals getting vaccination, that is, all susceptible individuals strictly move from S (or Spv) to E (Epv) compartments and, (ii) C+ and I+ (which also contribute to the transmission levels). The same argument is applied to equation 9. Please, clarify it.

3. It is not clear how the authors considered commuters. Equations 4 and 12 do not account for any fixed influx rate of those individuals. The current formulation as presented by the system of equations does not have any effect in the transmission dynamics if initial conditions for C+ and Cpv+ are zero. Please clarify how commuters were considered in the model dynamics and report the initial conditions for all model variables.

4. Equation 17 (dRi/dt) does not account for individuals receiving the second dose of the vaccine. Please, clarify it.

*It’s recommended that the authors provide the code used in this model.

Please, label and describe each panel of all multi-panel figures for better presentation and clarity. Figure 6 should be reformulated – the current form is difficult to understand. For better comparison between different scenarios, Figures 3, 5, 7 and 8 should have only one scale for Infected panels, as well as for ICU and Death panels. The scale for Figure 4 panels should also be standardized. The authors are reporting 50%CI instead of the commonly used 95%CI. If there is no strong justification for doing so, please report the 95%CI.

Why are the authors just presenting the age stratified results for the worst-case scenario (Fig 4)? Even considering the 50%CI, the model inferences go from around zero (difficult to visualize the minimum values) to a considerable increase in the number of cases for all ages. I understand the authors' caution, however, considering that the conclusions must be strictly based on the study's findings, this result does not support statements such as “we can safely assume that the number of infections will grow even faster after the end of the simulation by the end of August” or "if we go back to our pre-pandemic methods soon, these numbers are rapidly increasing".

The authors report that they obtained p* values from reference 11 (last sentence in page 5). I was not able to find values for pU and pD in the cited reference. Please, clarify it.

Equation 30 assumes that the Delta variant is 40% more infectious than the Alpha one, the text says 20-60% and the simulations considered 40 and 60%. Please, standardize and make it clear in the Methods section.

What do authors mean by “the number of individuals at the start of the simulations are extrapolated from the RKI-database and DVI-database” (Results – first paragraph)? Please, be specific about how and which data were obtained and from each database. This information should be added to the Methods section.

The authors should specify the incidence unit in the text. For example, in the passage “incidences of about 600”, do authors mean 600 cases/100.000 people? People, always specify it on the text.

What does exactly “correspond to Figure 2” as mentioned in the second line of page 12? Please, be clear about it.

**Have the authors made all data and (if applicable) computational code underlying the findings in their manuscript fully available?**

Reviewer #1: **No: **No code files provided

Reviewer #2: **No: **Data and code are not available.

PLOS authors have the option to publish the peer review history of their article (what does this mean?). If published, this will include your full peer review and any attached files.

Reviewer #1: No

Reviewer #2: No
---

## [Decision Letter · Decision Letter 1]

15 Dec 2021

Dear Mr. Koslow,

Thank you very much for submitting your manuscript "Appropriate relaxation of non-pharmaceutical interventions minimizes the risk of a resurgence in SARS-CoV-2 infections in spite of the Delta variant" for consideration at PLOS Computational Biology.

As with all papers reviewed by the journal, your manuscript was reviewed by members of the editorial board and by several independent reviewers. In light of the reviews (below this email), we would like to invite the resubmission of a significantly-revised version that takes into account the reviewers' comments.

We cannot make any decision about publication until we have seen the revised manuscript and your response to the reviewers' comments. Your revised manuscript is also likely to be sent to reviewers for further evaluation.

Sincerely,

Claudio José Struchiner, M.D., Sc.D.

Associate Editor

PLOS Computational Biology

Virginia Pitzer

Deputy Editor-in-Chief

PLOS Computational Biology

Reviewer's Responses to Questions

**Comments to the Authors:**

Reviewer #1: Dear Authors,

I appreciate the work to extend the model formulation and schematic. Thank you for letting me know about the nuances of using distributions in compartmental models.

As such, my concerns are suitably addressed and I am recommending this manuscript for publication.

Reviewer #2: PCOMPBIOL-D-21-01275R1: Appropriate relaxation of non-pharmaceutical interventions minimizes the risk of a resurgence in SARS-CoV-2 infections in spite of the Delta variant

The current version of the manuscript still has methodological issues that must be addressed for greater clarity of the work.

In the response to reviewers, with regards to the number of individuals at the start of the simulations being extrapolated from the RKI-database and DVI-database, the authors say: “This extrapolation is intuitive and a full description of the rather intuitive formulas would just extend the paper by two pages.” It may not be the best practice to rely on intuition and a brief description (or perhaps a supplemental material) would bring more clarity to the work. I respect the position of the authors thought.

Also in the response to reviewers, the authors mentioned they changed the term “0.5” (factor that multiplied the term (Cj+Cpv,j) in Equations 1,2,9,10,17,18) to “beta_Cj”. The current version of the manuscript presents “epsilon_Cj” and “epsilon_Ij”. The authors introduced these new parameters but did not present them in the text. They are only presented in Tables 1 and 2. What do they represent? What is the motivation to choose these forms (“sigmoidal curve from 0:5 to 1 on incidence 10 to 20” and “sigmoidal curve from [0:0; 0:2] to [0:4; 0:5] on incidence 10 to 150”)? How and from where were they derived?

Regarding the 95%CI, the authors’ response to reviewers is reasonable w.r.t the argumentation about the fact that “the 25 % or 75 % percentiles can provide valuable qualitative insight into possible scenarios exceeding the near future…”. Building on that, the authors should not consider, although clearly stated and justified, only the worst case for the simulated scenarios. For example, considering scenario 1, it is stated (line 365) that “early opening leads to a rise of infections from about 6-16 July onward.” This is the worst case shown in Fig 4. Even considering the 50%CI, we can infer from the model output that the cases can substantially rise or fall to something close to zero (hard to see), independently if the Delta variant is considered 40 or 60% more infectious than Alpha. The same argument for scenarios 2 and 3. In addition, graph scales chosen for Figs. 4, 5, 6 and 7 do not allow comparing model's predictions and the actual values during the first half of the simulation.

It is not clear to me why equations for vaccinated (17 and 18) and unvaccinated (1 and 2) individuals present the same parameter “pho_i”, while equations for partially vaccinated individuals (9 and 10) present a different value “pho_PV” (Tables 1 and 2 just present “pho_i_(0)”). Additionally, the authors say that certain parameters from equations 1-8 were reused in Equations 9-26 “without introducing new variable names” (lines 170-171). What does it mean? Which parameters? What was the reason for that? It is highly recommended that the description of the methods be as clear as possible.

Line 97: the authors define Zi := U_i=1_n{Si,Ei,Ci,…} as the set of all compartments of age group i. This definition of “Zi” seems to be, however, the set of all compartments of ALL age groups. At the same sentence, it is not comprehensible and not stated what “j” stands for in N_j_D.

An iterative method (page 12) was used to correct the differences between county of reported and home location of vaccination. Although the authors give a reference to the code, the method does not seem to be validated elsewhere. I wonder if it converges to a reasonable/correct solution and if its final results are independent of the order in which weights are corrected. Unfortunately, I cannot evaluate this point.

Equation 33 is different than in the previous version. In this current form, for any t between 0 and T_V_PV, D1<0. Please, clarify it.

The authors are referencing to Table 2 when, perhaps, it was supposed to refer to Table 1 (lines 115, 226, …)

The authors start page 14 arguing about the relative infectiousness of Delta w.r.t. Alpha and state: “60% is potentially closer to the true value”. In my opinion, this sentence is not under the scope of this work.

Line344: the number “7” is duplicated.

Line 353: “??”

**Have the authors made all data and (if applicable) computational code underlying the findings in their manuscript fully available?**

Reviewer #1: Yes

Reviewer #2: Yes

PLOS authors have the option to publish the peer review history of their article (what does this mean?). If published, this will include your full peer review and any attached files.

Reviewer #1: **Yes: **Affan Shoukat

Reviewer #2: No
---

## [Decision Letter · Decision Letter 2]

23 Mar 2022

Dear Dr. Kuehn,

We are pleased to inform you that your manuscript 'Appropriate relaxation of non-pharmaceutical interventions minimizes the risk of a resurgence in SARS-CoV-2 infections in spite of the Delta variant' has been provisionally accepted for publication in PLOS Computational Biology.

Before your manuscript can be formally accepted you will need to complete some formatting changes, which you will receive in a follow up email. A member of our team will be in touch with a set of requests. At that time, you may wish to address the very minor edits suggested by the reviewer below.

Best regards,

Claudio José Struchiner, M.D., Sc.D.

Associate Editor

PLOS Computational Biology

Virginia Pitzer

Deputy Editor-in-Chief

PLOS Computational Biology

Reviewer's Responses to Questions

**Comments to the Authors:**

Reviewer #2: PCOMPBIOL-D-21-01275R2: Appropriate relaxation of non-pharmaceutical interventions minimizes the risk of a resurgence in SARS-CoV-2 infections in spite of the Delta variant

My concerns were addressed. I would only point out that, in the supp material, the authors used the corresponding preprint of this same work as reference (ref 6). It would be advised to use the main text instead.

Main text

Lines 69 and 177: missing “.”.

Line 530: repetition of “of”.

**Have the authors made all data and (if applicable) computational code underlying the findings in their manuscript fully available?**

Reviewer #2: Yes

PLOS authors have the option to publish the peer review history of their article (what does this mean?). If published, this will include your full peer review and any attached files.

Reviewer #2: No

---

## [Editor Report · Acceptance letter]

3 May 2022

PCOMPBIOL-D-21-01275R2 

Appropriate relaxation of non-pharmaceutical interventions minimizes the risk of a resurgence in SARS-CoV-2 infections in spite of the Delta variant

Dear Dr Kühn,

I am pleased to inform you that your manuscript has been formally accepted for publication in PLOS Computational Biology. Your manuscript is now with our production department and you will be notified of the publication date in due course.

With kind regards,

Livia Horvath
